# A dynamic feedback loop between retrograde sterol transport and TORC2 controls adaptation of the plasma membrane to stress

Maria G Tettamanti [1,2], Paulina Nowak[1,2], Beata Kusmider [1], Jennifer M Kefauver [1,4],
Vincent Mercier[3], Aurélien Roux [2✉] & Robbie Loewith [1✉]

## Abstract

**Cells monitor and dynamically regulate the lipid composition and biophysical properties of their plasma membrane (PM). The Target Of Rapamycin complex 2 (TORC2) is a protein kinase that acts as a central regulator of plasma membrane homeostasis, but the mechanisms by which it detects and reacts to membrane stresses are poorly understood. To address this knowledge gap, we characterized a family of amphiphilic molecules that physically perturb plasma membrane organization and in doing so inhibit TORC2 in yeast and mammalian cells. Using fluorescent reporters of various lipids in budding yeast, we show that exposure to these small molecules causes mobilization of PM ergosterol as well as inhibition of TORC2. TORC2 inhibition results in activation of the PM–ER sterol transporters Lam2 and Lam4 and the subsequent rapid removal of accessible ergosterol from the plasma membrane via PM–ER contact sites. This sequence of events, culminating in the reactivation of TORC2, is also observed with several other PM stresses, suggesting that TORC2 acts in a feedback loop to control active sterol levels at the plasma membrane to maintain its homeostasis.**

**Keywords** TOR Complex 2; Plasma Membrane Stress; Sterol Transport; Membrane Tension; Small Amphipaths
**Subject Categories** Membranes & Trafficking; Signal Transduction

## Introduction

The plasma membrane (PM) is an interface between the cell and its surroundings. It serves both as a selectively permeable barrier and as a signaling platform (Leonard et al, 2023). The involvement of the PM in a large collection of cellular processes is enabled by its lateral compartmentalization into functional domains (Honigmann and Pralle, 2016; Lu and Fairn, 2018). This is dependent on the biophysical properties of the PM,

and thus on its complex protein and lipid composition and configuration. To maintain optimal PM properties, cells actively regulate their lipid repertoire (van Meer et al, 2008; Harayama and Riezman, 2018). Sterols, for example, are major modulators of physical membrane properties, such as rigidity and phase behavior, and they are especially enriched at the PM (Dufourc, 2008; Shaw et al, 2021; Doole et al, 2022). Recent studies have demonstrated that through their ability to readily flip-flop, sterols are able to buffer leaflet stresses (Varma and Deserno, 2022; Doktorova et al, 2025). As important factors in membrane microdomain organization (Kefauver et al, 2024), most sterol molecules in membranes are cloistered by phospho- and sphingolipids, and only the excess fraction is free and accessible for biochemical processes (Radhakrishnan and McConnell, 2000; Ohvo-Rekilä et al, 2002; Sokolov and Radhakrishnan, 2010; Lange et al, 2013; Das et al, 2014; Lange and Steck, 2020). Sterol levels in internal membranes vary and require tight regulation (van Meer et al, 2008). The redistribution of sterols between cellular membranes depends on various vesicular and non-vesicular transport routes, of which cytosolic and membrane-contact-site lipid transport proteins are the most efficient (Kaplan and Simoni, 1985; Baumann et al, 2005; Dittman and Menon, 2017; Wong et al, 2019; Steck et al, 2021).

Target Of Rapamycin Complex 2 (TORC2) is one of two functionally and structurally conserved protein complexes containing the essential protein kinase TOR (Heitman et al, 1991; Loewith et al, 2002; Sarbassov et al, 2004; Jacinto et al, 2004; Wullschleger et al, 2006). While specific inhibition by rapamycin has facilitated characterization of TORC1, TORC2 is rapamycin-insensitive and thus its characterization has relatively lagged (Loewith et al, 2002; Gaubitz et al, 2015). Recent studies have positioned yeast TORC2 as a central regulator of PM homeostasis: the application of various orthogonal stimuli that cause an increase in membrane tension - i.e., hypo-osmotic shock, sphingolipid biosynthesis inhibition - also cause an increase in TORC2 activity, while those that reduce membrane tension - hyper-osmotic shock and treatment with palmitoylcarnitine (PalmC) - rapidly inactivate TORC2 (Berchtold et al, 2012; Riggi et al, 2018). TORC2 regulates several processes that affect turgor pressure and other plasma membrane properties, via its primary effector, the AGC family kinase Ypk1 (reviewed in (Roelants et al, 2017; Thorner, 2022)). The retrograde PM–ER sterol transporters Lam2 and Lam4 are amongst the downstream

[1]Department of Molecular and Cellular Biology, University of Geneva, Geneva 1211, Switzerland. [2]Department of Biochemistry, University of Geneva, Geneva 1211, Switzerland. [3]ACCESS platform, Section of Chemistry of Biochemistry, University of Geneva, Geneva 1211, Switzerland. [4]Present address: Nanomaterials and Nanotechnology Research Center (CINN), Spanish National Research Council (CSIC), El Entrego, Spain. ✉E-mail: aurelien.roux@unige.ch; robbie.loewith@unige.ch

effectors regulated by Ypk1 (Roelants et al, 2018; Topolska et al, 2020), reinforcing TORC2's role in biophysical homeostasis of the PM. Consistent with this role, data generated with the lipid packing reporter dye Flipper-TR® suggest that acute chemical inhibition of TORC2 increases PM tension, while Ypk1 hyperactivation decreases it (Riggi et al, 2018). Collectively, these data have recently led us to propose that TORC2 functions in a mechanosensitive feedback loop to maintain the biophysical homeostasis of the PM (Riggi et al, 2020). This role of TORC2 is likely conserved in higher eukaryotes, as evidenced by reports showing that the major mTORC2 substrate Akt is regulated downstream of changes in membrane tension and that mTORC2 signaling feeds back to control PM properties (Kippenberger et al, 2005; Diz-Muñoz et al, 2016; Roffay et al, 2021; Ono et al, 2022).

How TORC2 monitors PM status and adapts signaling to react to different stresses remains poorly understood. In budding yeast, TORC2 stimulation by PM stress is dependent on the tension dependent release of the regulatory proteins Slm1 and Slm2 from the eisosomal compartment, a stress sensitive PM domain scaffolded by BAR domain proteins (Berchtold et al, 2012; Appadurai et al, 2020; Lanze et al, 2020; Sakata et al, 2022; Kefauver et al, 2024). Counterintuitively, TORC2 inhibition seems to be independent of Slm1/2 relocation and instead correlates with the formation of PI(4,5)P2-containing PM invaginations (Riggi et al, 2018). In neither case are the molecular mechanisms of TORC2 regulation well understood.

During hyperosmotic shock, transient TORC2 inhibition is paired with a transient activation of the Hog1 MAPK cascade. These changes in TORC2 and Hog1 activities converge on the production and efflux of the osmolyte glycerol to mitigate the offset of cell volume and turgor and ultimately restore homeostasis (Lee et al, 2012; Muir et al, 2015; Riggi et al, 2019; de Nadal and Posas, 2022). We previously identified the small amphiphile palmitoylcarnitine (PalmC) as an indirect TORC2 inhibitor (Riggi et al, 2018). PalmC inserts into the PM, and like hyperosmotic shock, reduces PM tension and transiently inhibits TORC2, but, interestingly, does not activate Hog1 (Riggi et al, 2018). This suggests that PalmC triggers a PM stress that is mechanistically distinct from hyperosmotic shock stress and likely requires different adaptive mechanisms to resolve. Building on this observation, we sought to use PalmC as a tool to better understand the physicochemical properties of the PM that are sensed upstream of TORC2. We found that exposure to small amphipaths causes a transient increase of free sterol molecules at the inner leaflet of the PM, which correlates with TORC2 inhibition. This in turn stimulates the rapid removal of free PM ergosterol, dependent on the TORC2-effectors and START-domain-containing proteins Lam2 and Lam4 (Gatta et al, 2015; Topolska et al, 2020; Murley et al, 2017), which is necessary for TORC2 reactivation. Subsequently, we found that this sterol mobilization and retrotranslocation is a common response to several PM perturbations and is similarly necessary for adaptation to these stresses, as readout by recovery of TORC2 activity. Thus, our data reveal a general, and likely conserved, mechanism of sterol mobilization and redistribution enabling cells to rapidly and dynamically respond to PM stress.

# Results

## PalmC partitions into the PM to inhibit TORC2

We have previously reported that the small amphipathic molecule PalmC (Fig. 1A, gray panel) interacts with the PM causing a decrease in membrane tension, the formation of PI(4,5)P$_2$-containing PM invaginations, and inhibition TORC2.

Mass spectrometry analysis of extracts from yeast cells treated with PalmC (in which PalmC is not a naturally occurring metabolite) showed that PalmC accumulates in the membrane fraction (Fig. 1B). Indeed, previous in vitro studies have revealed that PalmC virtually completely partitions into lipid bilayers in aqueous solution (Requero et al, 1995; Goñi et al, 1996). We observed that the kinetics and magnitude of PalmC effect correlated closely with the relative amount of PalmC to cell membranes rather than the concentration of PalmC per se (Fig. EV1A). This difference cannot be attributed to differences in culture stage, or secreted factors, since it is observed in cells from the same logarithmically growing culture, which were pelleted and resuspended to different culture densities (OD$_{600\,nm}$) in fresh media. Thus, controlling culture density offers a way to fine-tune PalmC effect at sub-CMC levels. We conclude that the effect of PalmC must be a consequence of its propensity to partition directly into cell membranes, in our case, most probably the plasma membrane.

## A family of small amphipaths with similar properties recapitulates the effect of PalmC

To characterize which molecular features of PalmC are necessary for its effect on the PM and TORC2, we performed a Structure Activity Relationship (SAR) analysis. We screened a panel of derivatives, differing from PalmC in either the length (or saturation) of their fatty acyl chain, or in their headgroup moiety (Fig. 1A), for their ability to recapitulate the PalmC effect.

First, we tested if any of these derivatives were able to cause PM invaginations both in live yeast cells expressing the PM PI(4,5)P$_2$ FLARE (fluorescent lipid-associated reporter) GFP-2xPH$^{PLC\delta}$ (Kavran et al, 1998) (Fig. 1C), and in a high-throughput microscopy screen with fixed cells expressing mCherry-2xPH$^{PLC\delta}$ (Fig. EV1B). Both approaches showed that a carbon tail of at least C14 was necessary to induce PM invaginations, and the effect was more substantial and lasted longer in carnitine derivatives with a carbon tail length of C16 or greater. In general, a cationic (choline, amine of long-chain bases) or zwitterionic (carnitine, PAF (as previously described (Kennedy et al, 2014)) headgroup was necessary for the induction of PM invaginations, while amphipaths with uncharged headgroups had no effect. We did not control for differences in PM insertion efficiency.

Next, we checked the effect of our derivatives on TORC2 activity. Substances that caused PM invaginations also inhibited TORC2-dependent phosphorylation of Ypk1 (Fig. 1D) (but not TORC1 phosphorylation of Sch9 (Fig. EV1C)). The inhibitory effect correlated with the size and duration of PM invaginations (Figs. 1C and EV1B). The inhibition of TORC2 by small amphipaths is conserved in mammalian cells, where PalmC (in a dose-dependent fashion) and derivatives that were effective in yeast trigger the loss of mTORC2-mediated phosphorylation of Akt (Fig. EV1D,E). We continued to characterize the effects of small amphipaths in budding yeast.

## The effects of small amphipaths are independent of metabolization

In our SAR screen, we had also included the long-chain sphingoid bases (LCBs) dihydrosphingosine (DHS) and phytosphingosine

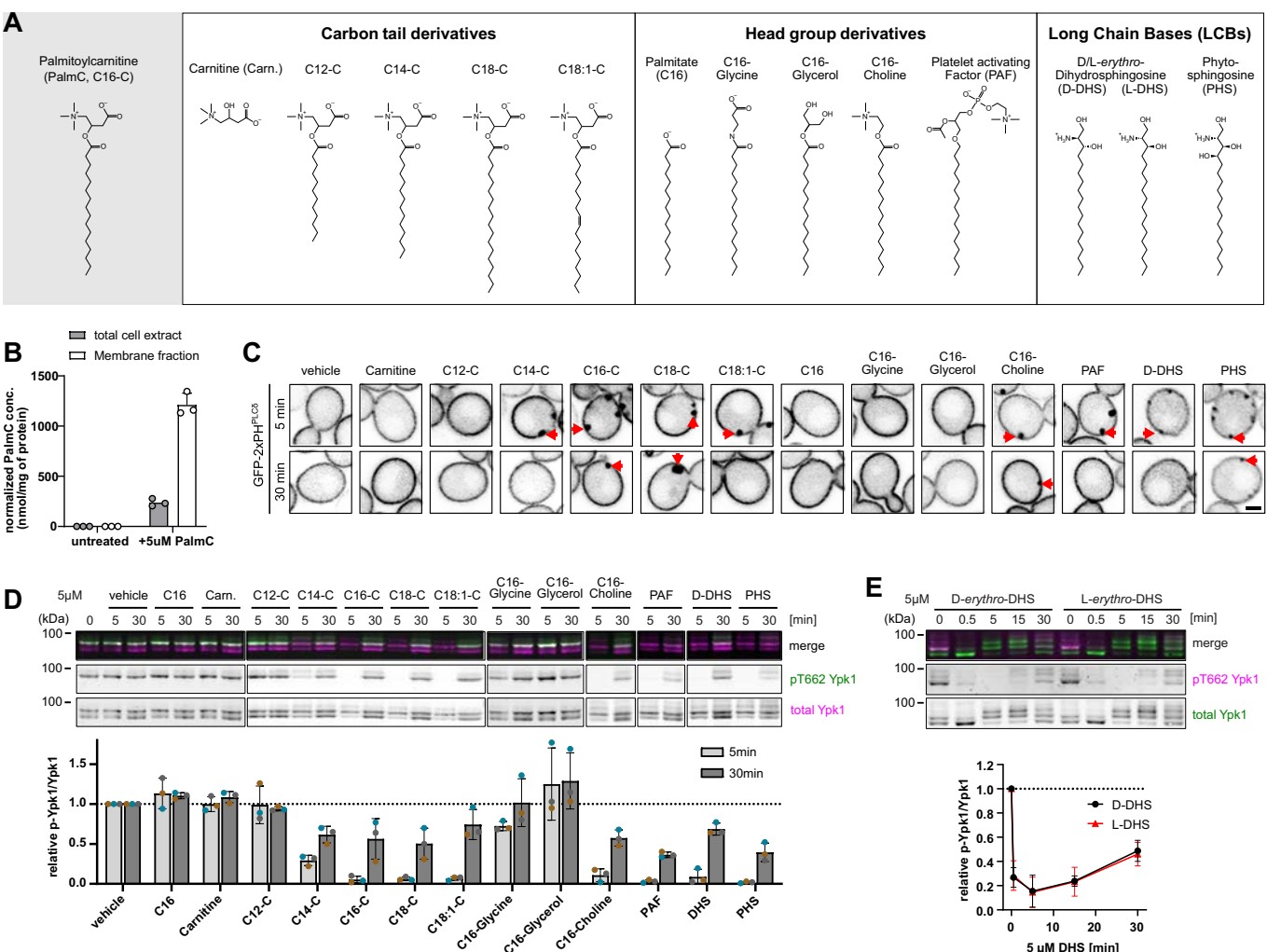

**Figure 1. A family of small amphipaths targets the plasma membrane to inhibit TORC2.**

(A) Structure formulas of tested PalmC derivatives. (B) Mass spectrometry analysis of whole cell and membrane fraction metabolite extracts from untreated yeast cells or yeast cells treated with 5 μM PalmC. The amount of PalmC was normalized to the protein content of each extract. Mean and SD of $N = 3$ independent experiments. (C) Live-cell fluorescence microscopy of WT cells expressing GFP-2xPH$^{PLC\delta}$ from plasmid. Cells were treated with the indicated substances at 5 μM for 5 or 30 min. Red arrows indicate PM invaginations. Scale bar = 2 μm. (D, E) Western blot analysis of TORC2 activity in WT yeast cells. Cells were treated (D) with 5 μM of different PalmC derivatives, or (E) with 5 μM D-erythro- or L-erythro- DHS for the indicated amounts of time, and TORC2 activity was assessed by relative phosphorylation of Ypk1. Mean and SD of $N \geq 3$ independent experiments. Source data are available online for this figure.

(PHS), which resemble PalmC in structure and effect, but unlike PalmC are metabolites naturally found in yeast. It had been reported previously that the addition of PHS leads to TORC2 inhibition (Lucena et al, 2018); however, in this previous work, it was postulated that processing of long-chain base into more complex sphingolipid species was required for TORC2 inhibition, a model at odds with our proposition that these amphiphiles act directly on the PM to trigger TORC2 inhibition. To challenge these models, we exploited a non-physiological/non-metabolizable enantiomer of DHS – L-erythro-DHS (Watanabe et al, 2002) (Fig. 1A, right panel). The addition of L-erythro-DHS triggered rapid (after 30 s) TORC2 inhibition with kinetics that were indistinguishable from D-erythro-DHS (Fig. 1E) and induced PM invaginations (Fig. EV1B). Thus, LCBs (and likely other amphipaths) directly act on the PM and inhibit TORC2 without the need to be metabolized.

For the remainder of the study, PalmC was used as a tool to induce PM perturbation, since it gave us the most robust effect in the SAR screen (see Figs. 1C,D and EV1B).

## Small amphipaths cause selective redistribution of PM ergosterol

Perturbation by PalmC results in PM remodeling, with 2xPH$^{PLC\delta}$ accumulating transiently in large PM invaginations (Riggi et al, 2018). To characterize better how PalmC affects the distribution of different lipid species, we complemented our assessments of PI(4,5) P$_2$ behavior (mCherry- or GFP-2xPH$^{PLC\delta}$), by additionally observing phosphatidylserine (PS; GFP-C2$^{Lact}$ (Yeung et al, 2008)), PI(4) P (mCherry-P4C (Luo et al, 2015)), and ergosterol (yeGFP-D4H (Maekawa et al, 2016; Marek et al, 2020)). We saw that both GFP-

C2$^{Lact}$ and mCherry-P4C similarly localized to the same PM invaginations after PalmC addition, where they colocalized with 2xPH$^{PLC\delta}$ (Figs. 2A and EV2A, left and middle panels, Fig. EV2B).

The ergosterol FLARE yeGFP-D4H however displays a strikingly different behavior: PalmC treatment leads to a significant loss of yeGFP-D4H signal at the PM, evidenced by a decrease in colocalization with mCherry-2xPH$^{PLC\delta}$ Figs. 2A and EV2A (right panels). D4H binds free sterols, i.e., the fraction of unesterified sterols inside membranes that is not in complexes with phospholipids or proteins (Johnson et al, 2012; Maekawa and Fairn, 2015). The yeGFP-D4H signal in exponentially growing budding yeast is heterogenous but often enriched in buds and at bud necks (Encinar Del Dedo et al, 2021) (see Fig. EV2C, top panel). Addition of vehicle has no impact on this pattern (Figs. 2B and EV2D; Movie EV1). Upon PalmC addition, yeGFP-D4H can initially be seen in PM invaginations, similarly enriched as mCherry-2xPH$^{PLC\delta}$ (Fig. EV2B), however, it rapidly dissociates from the PM, eventually showing diffuse cytoplasmic signal and bright intracellular foci (Fig. 2B; Movie EV2). During early timepoints, intracellular foci are usually in close vicinity to ER (Fig. EV2E). They do not colocalize with early endosomes (marked with FM4-64 or GFP-Vps21; Fig. EV2F,G) or lipid droplets (stained with Lipidspot$^{TM}$ 488; Fig. EV2H). However, we observe an increase in cellular lipid droplet volume following PalmC treatment (Fig. EV2I). In line with our previous observations, two PalmC derivatives (C16-Choline and PHS) which caused PM invaginations and TORC2 inhibition also induced relocation of free PM ergosterol, while an ineffective derivative (C16-Glycerol) did not (Fig. EV2J). We conclude that PM ergosterol relocation is a consequence of the PM perturbing properties of small amphipaths.

## PalmC-induced sterol internalization is Lam2/4 dependent

Internalization of PM sterol (and GFP-D4H) to intracellular foci has been previously described in fission yeast treated with the Arp inhibitor CK-666, where it was dependent on a StART (steroidogenic acute regulatory protein-related lipid transfer) family sterol transporter (Marek et al, 2020). In budding yeast, TORC2 has been reported to both regulate (Murley et al, 2017) and be regulated by the PM–ER-contact site resident proteins Lam2 and Lam4 (Roelants et al, 2018; Topolska et al, 2020). Lam2/4 contain two START domains, of which at least one has been demonstrated to facilitate sterol transport between membranes; in vivo, Lam2/4 seems to predominantly transport sterols from the PM to the ER, following the concentration gradient (Gatta et al, 2015; Jentsch et al, 2018; Tong et al, 2018). Indeed, in cells expressing GFP-Lam2, we observed that PalmC-induced PM invaginations often formed at sites with preexisting GFP-Lam2 foci (Fig. EV2K, cyan arrow), although GFP-Lam2 foci did not always colocalize with invaginations (Fig. EV2K, yellow arrow) and vice versa. To test if sterol internalization upon PalmC treatment is dependent on Lam2/4, we observed sterol behavior in lam2Δ lam4Δ cells. Virtually all mutant cells showed a strong yeGFP-D4H signal at the PM, in agreement with previous work that showed an excess of free PM ergosterol (Fig. EV2C, middle panel) (Gatta et al, 2015). Application of PalmC to lam2Δ lam4Δ cells triggered the formation of large PM invaginations. In contrast to WT however, we observed that sterols are not removed from the PM and instead yeGFP-D4H continued

to colocalize with mCherry-2xPH$^{PLC\delta}$, both inside and outside of PM invaginations (Figs. 2C and EV2L; Movies EV3 and EV4). This confirms that PalmC-induced sterol internalization from the PM is dependent on Lam2/4.

## Lam2/4-mediated internalization of PM sterols affects TORC2 activity after PalmC

To understand how these changes in sterol behavior affect TORC2 regulation, we monitored Ypk1 phosphorylation. Using a phosphospecific antibody, we did not observe an increase in baseline TORC2 activity in lam2Δ lam4Δ cells, which had been previously reported by electrophoretic mobility shift (Murley et al, 2017). Instead, baseline TORC2 activity was consistently slightly decreased in these cells (Fig. 2D). Ypk1, activated directly by TORC2, inhibits Lam2 and Lam4 through phosphorylation on Thr518 and Ser401, respectively (Roelants et al, 2018; Topolska et al, 2020). We substituted these residues with alanine, generating a strain in which Lam2/4 were no longer inhibited by phosphorylation (Roelants et al, 2018). In these cells, yeGFP-D4H showed that free sterols were constitutively shifted away from the PM to intracellular structures (Fig. EV2C, bottom panel). Intriguingly, in opposition to lam2Δ lam4Δ cells, basal TORC2 activity was increased in LAM2$^{T518A}$ LAM4$^{S401A}$ cells (Fig. 2D). This suggests that a decrease in free PM sterols stimulates TORC2 activity, in which case, mild sterol depletion should have a similar effect. To test this, we observed growth and TORC2 activity while treating cells with the ergosterol synthesis inhibitors Atorvastatin and Fluconazole (Jordá and Puig, 2020) (Fig. EV2M). Indeed, before the appearance of growth inhibition, both drugs induced an increase in TORC2 activity. However, after the appearance of growth inhibition, TORC2 activity is decreased, likely because further sterol depletion compromises PM integrity.

We further wondered if Lam2/4-mediated PM sterol extraction is necessary for TORC2 activity recovery from PalmC. Indeed, while upon PalmC treatment, TORC2 is rapidly inhibited in both WT and lam2Δ lam4Δ cells, TORC2 recovery is strongly delayed in lam2Δ lam4Δ cells (Fig. 2E, top and middle blots). In LAM2$^{T518A}$ LAM4$^{S401A}$ cells, TORC2 activity recovers with similar kinetics as the WT (Fig. 2D, bottom blot), suggesting that Lam2/4 release from TORC2-dependent inhibition during PalmC treatment is a fast and efficient process in WT cells, not further expedited by these constitutively active Lams. We conclude that Lam2/4-mediated PM sterol extraction stimulates TORC2 activity and mediates its recovery after PalmC.

## TORC2 inhibition triggers retrograde transport of PM ergosterol

To pinpoint the specific role of TORC2 activity regulation for sterol internalization during PalmC treatment, we inhibited TORC2 pharmacologically. In budding yeast, it is possible to render TORC2 sensitive to rapamycin by truncating a C-terminal part of the TORC2 subunit Avo3, and TORC1 insensitive to rapamycin by expressing the TOR1-1 allele (Gaubitz et al, 2015). Addition of rapamycin, but not vehicle, to AVO3-ΔCT TOR1-1 cells induced the relocation of yeGFP-D4H to internal foci, but only after around 30 min (Figs. 3A and EV3B; Movies EV5 and EV6). Even considering that TORC2 inhibition by rapamycin is slower than

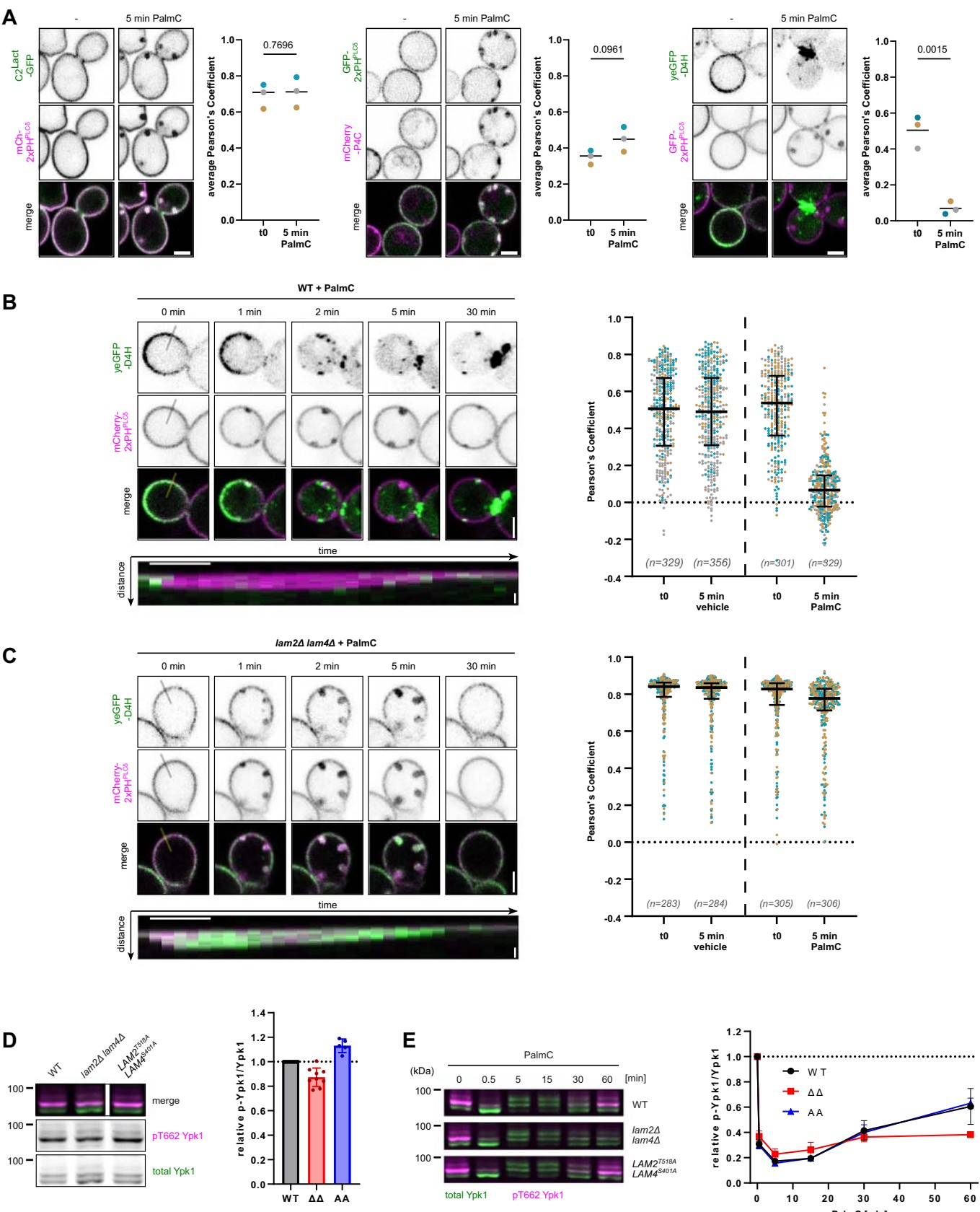

Figure 2. Lam2/4-mediated PM-ergosterol redistribution is required for TORC2 recovery after PalmC.

(A) Live-cell fluorescence microscopy of yeast cells expressing a PI(4,5)P2 reporter (GFP- or mCh-2xPH$^{PLC\delta}$) along with a phosphatidylserine (PS) reporter (C2$^{Lact}$-GFP), a PI4P reporter (mCh-P4C), or a free ergosterol reporter (yeGFP-D4H). Cells were treated with 5 µM PalmC for 5 min, and the colocalization of each reporter with 2xPH$^{PLC\delta}$ was quantified (Pearson's correlation coefficient). Representative images and quantifications (average single-cell values, mean, P value determined by unpaired t test) are shown. Different colors represent data from N = 3 independent experiments. Scale bar = 2 µm. (B, C) Live cell fluorescence microscopy of free ergosterol (yeGFP-D4H) and PI(4,5)P2 (mCh-2xPH$^{PLC\delta}$) before and after the addition of vehicle or 5 µM PalmC in (B) WT and (C) lam2Δ lam4Δ cells. Time-lapse images show the localization of yeGFP-D4H relative to mCh-2xPH$^{PLC\delta}$ at the indicated timepoints. Scale bar = 2 µm. Kymographs (bottom panels) depict yeGFP-D4H distribution relative to a mCh-2xPH$^{PLC\delta}$-marked PM invagination along the specified line over 30 min, at 1-min intervals. Scale bars: x = 5 min, y = 0.5 µm. Scatter plots show the colocalization between the two probes before and after 5 min of treatment, data points represent individual cells, plotted with median and interquartile range. Different colors represent data from independent experiments. (D, E) Western blot analysis of TORC2 activity in WT or the indicated mutants (D) at steady state or (E) before and after treatment with 5 µM PalmC, as assessed by relative phosphorylation of Ypk1. Mean and SD of N ≥ 3 independent experiments. Source data are available online for this figure.

during PalmC treatment (Gaubitz et al, 2015), these sterol kinetics are different from the ones observed after PalmC treatment by an order of magnitude. Thus, while TORC2 activity is clearly necessary to maintain PM ergosterol levels, TORC2 inhibition, and thereby Lam2/4 activation, cannot be the only trigger for PalmC-induced rapid sterol internalization. This is consistent with our previous findings that non-phosphorylable mutants of Lam2/4 do not differ from WT in the recovery rate of TORC2 activity after PalmC.

## PalmC transiently increases biochemically accessible ergosterol at the PM

It has been previously shown that small amphipathic molecules render cholesterol molecules in membranes more accessible (Lange et al, 2009), presumably by displacing them from microdomains. Moreover, an excess of lipids in the outer leaflet of the PM was recently demonstrated to trigger sterol flipping to the inner leaflet followed by retrograde transport through the ER to lipid droplets in mammalian cells (Doktorova et al, 2025). Thus, we speculated that PalmC integration into the PM might directly stimulate retrograde ergosterol transport by Lam2/4 by increasing the PM pool of free ergosterol at the inner PM leaflet. Indeed, we could observe that immediately after the addition of PalmC, the PM signal of yeGFP-D4H increased in some cells (Fig. EV3A). However, this effect was clearly visible only in cells that had little to no yeGFP-D4H at the PM before treatment (and thus a mCherry-2xPH$^{PLC\delta}$ colocalization coefficient around 0). Only a very small fraction of cells falls into this category in steady-state conditions (see Fig. 2B), rendering this very transient effect difficult to measure across a population in WT cells. To address this limitation, we opted to use rapamycin (30 min)-pretreated AVO3-ΔCT TOR1-1 cells, where PM yeGFP-D4H localization is drastically reduced across the population without direct perturbation of the PM. Addition of PalmC to these cells caused a small but noticeable transient increase of yeGFP-D4H signal at the PM, showing a release of a pool of previously inaccessible ergosterol, which was afterwards rapidly internalized (Figs. 3B and EV3B,C). PalmC addition to LAM2$^{T518A}$ LAM4$^{S401A}$ cells likewise resulted first in a transient increase and then a further decrease in PM yeGFP-D4H signal (Figs. 3C and EV3D). These results confirm that perturbation of the PM by PalmC leads to an increase of biochemically available ergosterol at the inner PM leaflet.

## Hyperosmotic shock recovery of PM and TORC2 is partially Lam2/4 dependent

To better understand the role of sterol redistribution in the PM stress response, we decided to extend our observations to additional membrane stresses. Hyperosmotic shock, like PalmC, causes a decrease in membrane tension and TORC2 inhibition (Riggi et al, 2018). Since artificial extraction of sterols from membranes has previously been shown to increase membrane tension (Biswas et al, 2019; Cox et al, 2021), we wondered if this free sterol retrograde transport is a general response that serves to restore membrane tension by removing lipids—and thus membrane area—from the PM. Unlike PalmC, hyperosmotic shock doesn't directly alter PM composition by the addition of exogenous lipid. If sterol internalization from the PM is triggered by a decrease in membrane tension and serves to alleviate this stress, we would expect that TORC2 recovery is also dependent on sterol internalization during hyperosmotic shock.

Both in WT and in lam2Δ lam4Δ cells, yeGFP-D4H formed clusters at the PM upon hyperosmotic shock with 1 M sorbitol, which often overlapped with the shallow PM invaginations observed with mCherry-PH$^{PLC\delta}$. In lam2Δ lam4Δ cells, PM invaginations were deeper, and some took longer to resolve. While the majority of yeGFP-D4H signal remained at the PM, after around 10-15 min we observed a slow internalization of a minor fraction of yeGFP-D4H signal to internal foci in WT, but not in lam2Δ lam4Δ cells (Fig. 4A; Movies EV7 and EV8). Under normal conditions, we could not observe an effect of LAM2/4 deletion on hyperosmotic shock recovery (Fig. EV4A). However, in hog1Δ cells, which are deficient in hyperosmotic shock adaptation, the additional deletion of LAM2/LAM4 further impaired TORC2 activity recovery (Fig. 4B). Thus, although much of the recovery from hyperosmotic shock is mediated by Hog1 signaling, PM ergosterol removal also contributes and may therefore be a general response to PM stresses.

## The adaptation of the PM to Arp2/3 inhibition is Lam2/4 dependent

As mentioned above, inhibition of the nucleators of branched actin filaments, Arp2/3, by CK-666 was previously shown to induce StART family sterol transporter-dependent PM sterol relocalization in S. pombe, albeit through an unknown mechanism (Marek et al, 2020). Given that inhibition of Arp2/3 blocks endocytosis, it seems plausible that CK-666 might lead to a loss of PM tension through an accumulation of excess membrane. We thus wondered if Arp2/3 inhibition might represent another condition in which sterol internalization serves to mitigate PM stress and if TORC2 might be implicated in this mitigation. We first confirmed that CK-666 causes Lam2/4-dependent PM sterol removal in S. cerevisiae. Indeed, CK-666 induced an internalization of yeGFP-D4H in WT cells, while

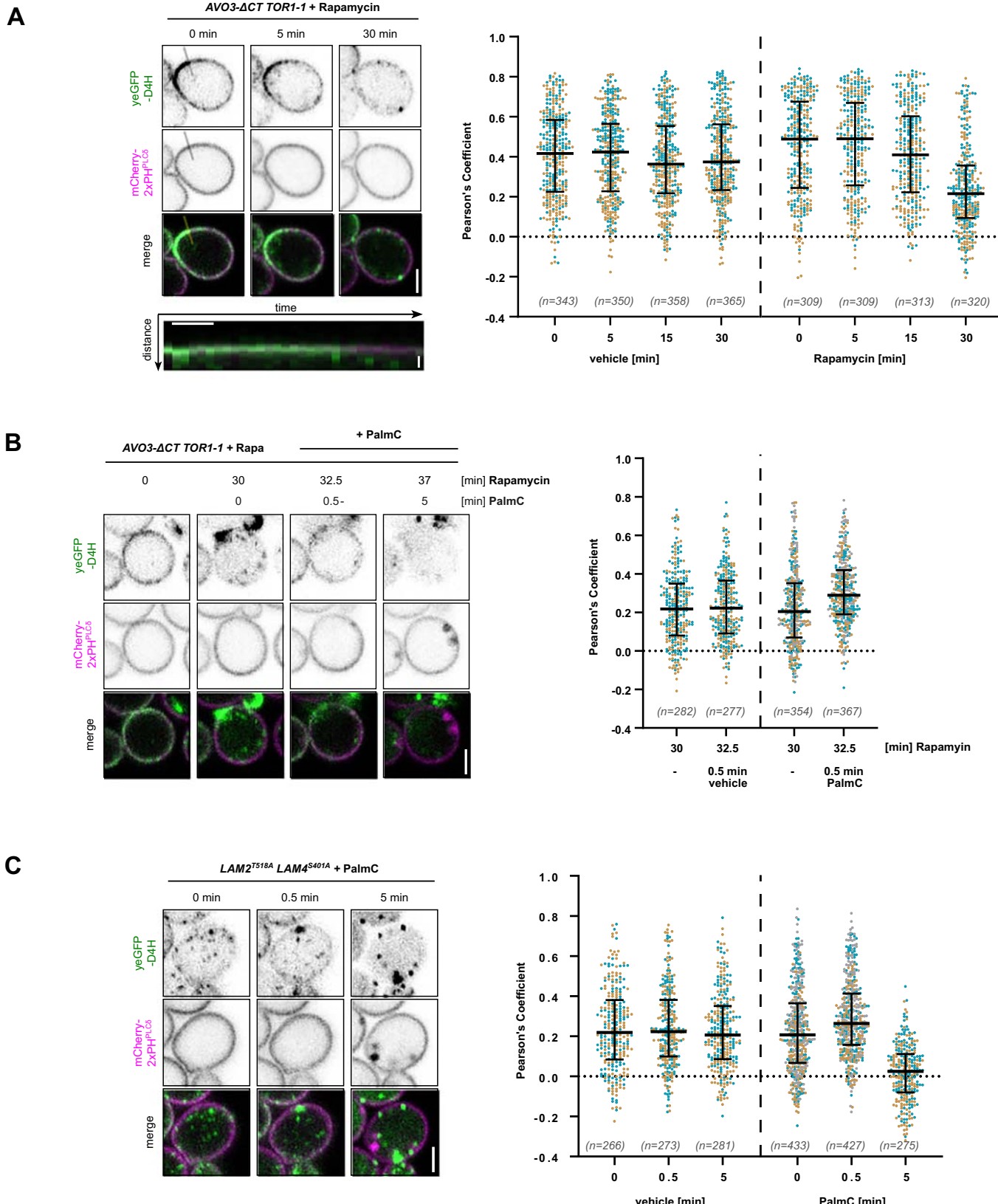

◀

**Figure 3. TORC2 inhibition and PalmC-induced increase in free sterols drive PM sterol internalization.**

(A–C) Live cell fluorescence microscopy of free ergosterol (yeGFP-D4H) and PI(4,5)P2 (mCh-2xPH^PLCδ) before and after the addition of (**A**) vehicle or 200 nM Rapamycin to *AVO3-ΔCT TOR1-1* cells, (**B**) 200 nM Rapamycin for 30 min, followed by 5 μM PalmC to *AVO3-ΔCT TOR1-1* cells (**C**) 5 μM PalmC to *LAM2^T518A LAM4^S401A* cells. Time-lapse images show the localization of yeGFP-D4H relative to mCh-2xPH^PLCδ at the indicated timepoints. Scale bar = 2 μM. Kymographs (bottom panels, if present) depict yeGFP-D4H distribution relative to a mCh-2xPH^PLCδ-marked PM invagination along the specified line over 30 min, at 1-min intervals. Scale bars: $x = 5$ min, $y = 0.5$ μm. Scatter plots show the colocalization between the two probes before and at the indicated timepoints post-treatment, data points represent individual cells, plotted with median and interquartile range. Different colors represent data from independent experiments. Source data are available online for this figure.

in *lam2Δ lam4Δ* cells it instead caused the appearance of transient PM invaginations, similar to the ones observed with PalmC (Figs. 4C, arrow and EV4B; Movies EV9 and EV10), and reminiscent of previous observations in *S. pombe* (Marek et al, 2020). Intriguingly, while CK-666 caused a slight decrease in TORC2 activity after 5 min in WT, this effect was much more prominent in *lam2Δ lam4Δ* cells (Fig. 4D). Thus, similarly to PalmC, Arp2/3 inhibition induces a TORC2-inhibiting PM stress which is mitigated by Lam2/4-dependent PM-sterol retrograde transport.

## The response of TORC2 to heat shock stress is partially Lam2/4 dependent

If the effect of small amphipaths is due to perturbation of sterol containing membrane domains, TORC2 should also be regulated in a Lam2/4 dependent way during heat shock, which leads to dissolution of lipid domains (Beney and Gervais, 2001; Los and Murata, 2004; Baumgart et al, 2007; Blicher et al, 2009; Edwards et al, 2016; Burns et al, 2017; Leveille et al, 2022) To assess this, we first observed PM sterols with yeGFP-D4H after inducing a rapid temperature change in our sample with a thermalizing chip. Heat shock induced a rapid removal of free sterols from the PM to internal compartments, and a clear decrease in the PM-to-cytoplasm ratio of yeGFP-D4H intensity in WT, but not in *lam2Δ lam4Δ* cells (Figs. 4E and EV4C; Movies EV11 and EV12). This treatment caused a decrease of mCherry-2xPH^PLCδ signal at the PM, especially in *lam2Δ lam4Δ* cells; however, we could still observe the formation of what appeared to be transient PM invaginations in these cells during heat shock (Movie EV12). To observe TORC2 activity during heat shock, WT and *lam2Δ lam4Δ* cells were grown at 25 °C, and small culture volumes were shifted to 37 °C. In WT cells, this elicited a rapid transient TORC2-activity spike but in *lam2Δ lam4Δ* cells a transient decrease in TORC2 activity was observed (Fig. 4F).

We conclude that heat shock triggers both a signal that transiently activates TORC2, and in parallel, sterol mobilization associated with TORC2 inhibition. In *lam2Δ lam4Δ* cells, where mobilized sterols cannot be cleared from the PM, the inhibitory signal dominates. Although further investigation into the mechanisms governing the role of TORC2 in heat shock recovery is necessary, from this result, we can conclude that heat shock represents another PM perturbing stress in which TORC2 activity is increased in a Lam2/4-dependent way.

## Discussion

We have identified a family of small amphipathic molecules that intercalate into the plasma membrane, causing large PM

invaginations and TORC2 inhibition, similarly to PalmC treatment (Riggi et al, 2018). Further characterization revealed that small amphipath-induced PM perturbation leads to rapid ergosterol mobilization and subsequent Lam2/4-dependent retrograde transport out of the PM, which is important for the recovery of TORC2 activity. Qualitatively similar events are observed in response to distinct PM stresses (hyperosmotic shock, block of endocytosis with CK-666, and heat shock), suggesting that these represent a general adaptation mechanism (Fig. 5A).

### Small amphipaths as membrane perturbing tools

PalmC has been previously shown to cause a perturbation of PM biophysical properties, i.e., PM tension and lipid order. The observation that PalmC also similarly affects ATP-depleted cells and giant unilamellar vesicles (GUVs) suggests that it acts passively by direct intercalation into the PM (Riggi et al, 2018). Our results corroborate this notion: the fact that the effect of small amphipaths is determined by the total dose per cell suggests that large amounts need to intercalate into the PM to physically perturb it (Fig. 5), and our SAR screen showed that the effect of PalmC is a consequence of its physicochemical properties rather than specific chemistry. Derivatives with longer carbon chains, and thus increased hydrophobicity and membrane partitioning (Requero et al, 1995; Ho et al, 2002) caused stronger effects. This confirms that their primary action is to insert into the PM, as opposed to direct binding to TORC2. Furthermore, comparison of the naturally occurring sphingolipid precursor D-*erythro*-DHS and the non-metabolizable L-*erythro*-DHS (Watanabe et al, 2002) showed that their effect as TORC2-inhibitors is virtually indistinguishable. This observation strongly speaks against a metabolic component of the rapid effects of PalmC. Our observations with the D4H probe agree with an in vitro study showing that small amphipaths mobilize sterols from lipid complexes (Lange et al, 2009). Our findings could extend to physiological or pharmacological molecules with similar properties to PalmC. For instance, alkylphospholipid analogs (Fei et al, 2023) have been shown to downregulate TORC2 signaling (Gills and Dennis, 2009; Nomura and Inoue, 2024) and relocalize sterols in budding yeast (Zaremberg et al, 2005).

### How is TORC2 regulated by membrane stress?

Using small amphipaths as a tool to perturb the PM, we found that their intercalation causes the mobilization and subsequent retrograde transport of PM sterols, and the appearance of large PM invaginations, both temporally correlating with TORC2 inhibition. We (Riggi et al, 2018), and others (Singer-Krüger et al, 1998; Stefan et al, 2002; Walther et al, 2006; Rodríguez-Escudero et al, 2018; Sakata et al, 2022; Phan et al, 2025), have associated PI(4,5)P₂ to

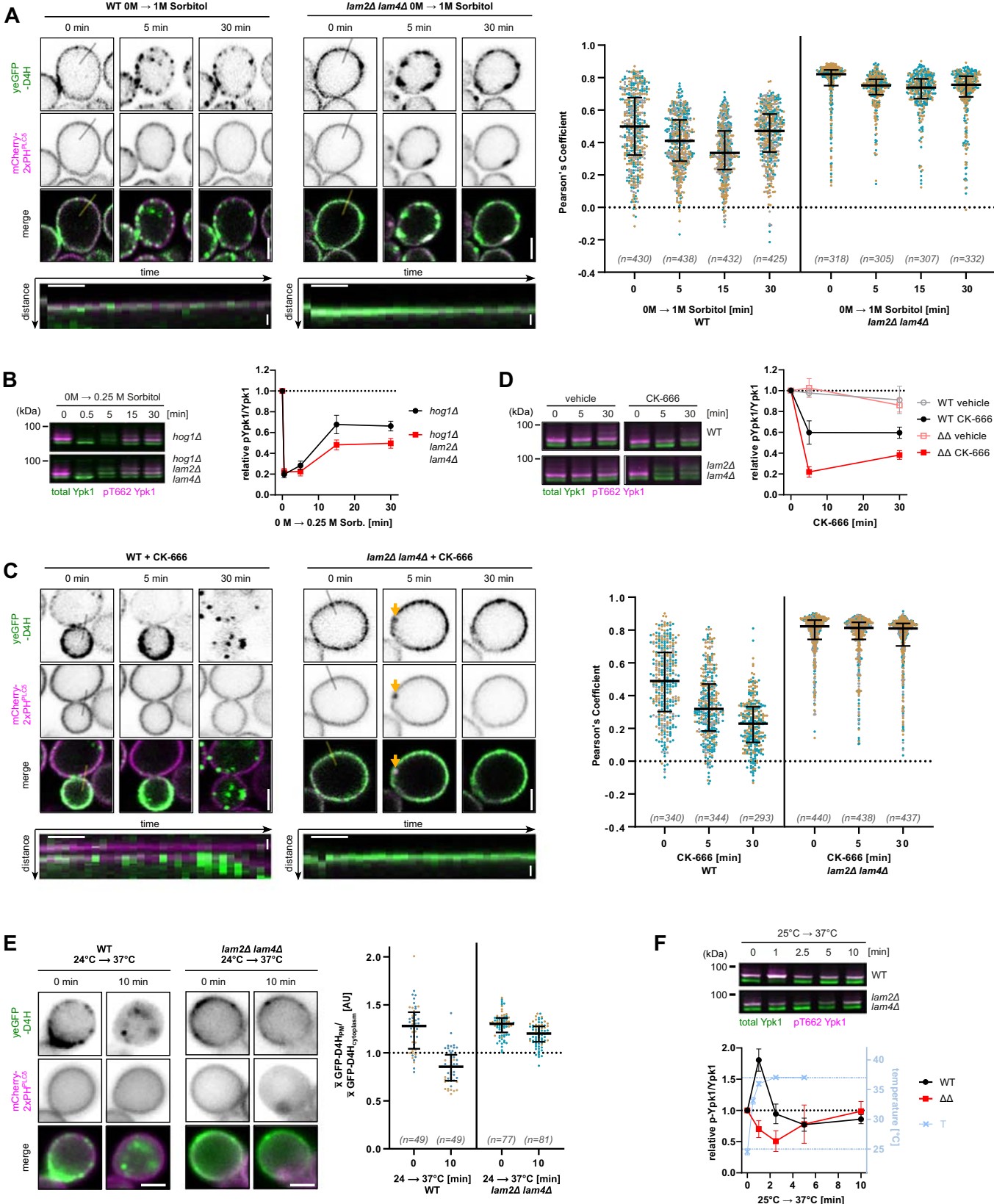

**Figure 4.   PM adaptation and TORC2 response to several stresses are partially Lam2/4 dependent.**

(A, C, E) Live cell fluorescence microscopy of free ergosterol (yeGFP-D4H) and PI(4,5)P2 (mCh-2xPH$^{PLC\delta}$) in WT or *lam2Δ lam4Δ* cells before and after (A) the addition of 1 M sorbitol media, (C) the addition of 250 μM CK-666, E) the application of a 24 °C = > 37 °C temperature shift. Time-lapse images show the localization of yeGFP-D4H relative to mCh-2xPH$^{PLC\delta}$ at the indicated timepoints. Scale bar = 2 μm. Kymographs (bottom panels, if present) depict yeGFP-D4H distribution relative to a mCh-2xPH$^{PLC\delta}$-marked PM invagination along the specified line over 30 min, at 1-min intervals. Scale bars: *x* = 5 min, *y* = 0.5 μm. Scatter plots show (A, C) the colocalization between the two probes before and at the indicated timepoints post-treatment or (E) the relative enrichment of GFP-D4H at the PM over cytoplasm. Data points represent individual cells, plotted with median and interquartile range. Different colors represent data from independent experiments. (B, D, F) Western blot analysis of TORC2 activity in (B) *hog1Δ* or *hog1Δ lam2Δ lam4Δ* cells after treatment with 0.5 M sorbitol, or WT or *lam2Δ lam4Δ* cells after (D) treatment with 250 μM CK-666 or vehicle (F) application of a 24 °C = > 37 °C temperature shift, as assessed by relative phosphorylation of Ypk1 (left *y* axis). Temperature measurements are plotted on the right *y* axis. Mean and SD of *N* ≥ 3 independent experiments. Source data are available online for this figure.

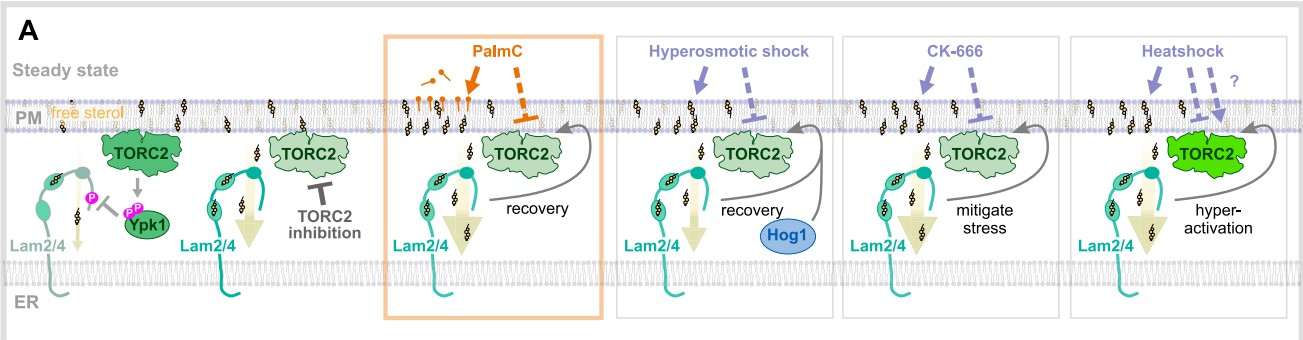

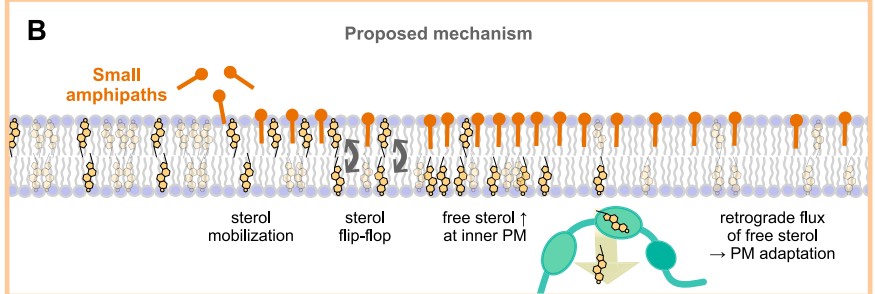

**Figure 5.   TORC2-regulated PM sterol retrograde transport as a general mechanism to adapt to PM stresses.**

(A) During steady state, the majority of PM ergosterol is biochemically inaccessible. An excess of free sterols or TORC2 inhibition, and thereby release of Lam2/4 inhibition, promotes PM–ER retrograde transport of sterols. Several stresses (small amphipaths, hyperosmotic shock, branched actin nucleation inhibition by CK-666, heat shock) lead to an increase in sterol retrograde transport by Lam2/4 and TORC2 inhibition. For simplicity, the model does not depict the PM invaginations that are also triggered by these stresses. Retrograde sterol transport has a role in relieving the PM stress and adapting the PM to a post-stress status, thus reactivating TORC2. In the case of hyperosmotic shock, the Hog1 pathway has a major role in stress recovery. In the case of heat shock, there appears to be a second signal that activates TORC2. (B) Proposed speculative mechanism for small amphipaths: large amounts of small amphipaths intercalate passively into the outer leaflet PM, which leads to the mobilization of sterols from lipid complexes. Free sterols flip-flop to the inner PM leaflet to buffer differential stress. At the inner PM leaflet, excess free sterols are extracted by disinhibited Lam2/4 sterol transporters, which adapt the PM to its post-stress status.

formation of these PM invaginations in yeast, and it has been proposed that their formation may serve to absorb excess membrane upon acute loss of PM tension. However, if and how these domains contribute to TORC2 inactivation is unclear. Our new observations with the D4H FLARE show that the formation of these invaginations after PalmC treatment is preceded by a rapid increase of free ergosterol at the PM.

Several observations make it tempting to speculate that mobilized sterols are, or are part of, the inhibitory signal to TORC2: (i) multiple PM stresses (PalmC, hyperosmotic shock, CK-666, heat shock) trigger sterol mobilization and TORC2 inactivation that is mitigated by sterol retrograde transport (Fig. 5), (ii)

chemical inhibition of sterol biosynthesis and specific depletion of free ergosterol at the PM (in *LAM2$^{T518A}$ LAM4$^{S401A}$* cells) results in TORC2 hyperactivation; while, (iii) blocking sterol retrograde transport—thus increasing free PM sterol levels—results in reduced TORC2 activity. We can so far not exclude the possibility that sterols act on TORC2 directly; however, we consider it more likely that sterols act indirectly by modulating PM parameters, such as lipid phase separation and packing (Dufourc, 2008; Shaw et al, 2021; Doole et al, 2022; Varma and Deserno, 2022; Doktorova et al, 2025). Future studies, such as reconstitution of TORC2 kinase activity on liposomes, will be aimed at understanding the possible mechanistic link between mobilized sterols and TORC2 inhibition.

# PM sterol retrograde transport as a general mechanism to adapt to PM stresses

Sterols are important modulators of membrane biophysics, and recent studies have highlighted how their ability to rapidly flip-flop between bilayer leaflets can buffer differential tension stresses (Varma and Deserno, 2022; Doktorova et al, 2025). In the same study, it was also demonstrated that the addition of phospholipids to the outer PM leaflet causes an excess of free sterol at the inner PM leaflet, and its subsequent retrograde transport to lipid droplets (Doktorova et al, 2025). Although we cannot exclude that it is the substrate of a flippase or scramblase, PalmC is not a metabolite found in yeast, nor, given its charged headgroup, is it likely to spontaneously flip to the inner leaflet (Goñi et al, 1996). Thus, we propose that PalmC accumulates in the outer leaflet, where it frees a fraction of sequestered sterols due to its membrane perturbing properties (Lange et al, 2009; Requero et al, 1995), and disrupts the lipid balance with the inner leaflet, which is, similarly to the mammalian cell model (Doktorova et al, 2025), rectified by sterol flipping and internalization (Fig. 5B). Hyperosmotic shock and endocytosis inhibition (CK-666) might cause an increase in free PM sterol through a loss of PM tension, which reduces lipid phase separation (Ayuyan and Cohen, 2008; García-Arcos et al, 2025), while heat shock melts lipid domains (Beney and Gervais, 2001; Los and Murata, 2004; Baumgart et al, 2007; Blicher et al, 2009; Edwards et al, 2016; Burns et al, 2017; Leveille et al, 2022), which we speculate leads to an increase in free PM sterol (Fig. EV5). In conclusion, we propose that sterol mobilization out of nanodomains is a general consequence of perturbations that result in a perceived excess of PM such as loss of membrane tension, or lipid imbalance between the lipid bilayers. The excess fraction of free sterols is subsequently passively cleared from the PM by Lam2/4 and can ultimately be stored in lipid droplets. This sterol transport-dependent PM adaptation plays a greater or lesser role in recovery depending on the specific stress. Free sterols, or a biophysical consequence of free sterols (e.g., loss of membrane packing defects), appear to generate a cue that leads to TORC2 inactivation and consequent signaling events needed to restore the biophysical properties of the PM, including activation of Lam2/4. Future studies will be aimed at identifying the PM cue generated by free sterols and how it leads to TORC2 inhibition.

# Methods

### Reagents and tools table

| Reagent/resource | Reference or source | Identifier or catalog number |
|---|---|---|
| **Experimental models** | | |
| All *S. cerevisiae* strains | | |
| *MATa leu2-3,112 ura3-52 rme1 trp1 his3* | Loewith lab | TB50a |
| *MATα leu2-3,112 ura3-52 rme1 trp1 his3* | PMID:26028537 | TB50α |
| TB50a + pRS426-GFP-2xPH(PLCδ) | This study | MTY002 |
| TB50a URA3::pRS406-mCh-2xPH(PLCδ) | This study | JK006 |

| Reagent/resource | Reference or source | Identifier or catalog number |
|---|---|---|
| TB50a + pRS414-pTdh3-mCherry-2xPH(PLCδ) + pRS416-LactC2-GFP | This study | BK3 |
| TB50a + pRS426-GFP-2xPH(PLCδ) + pRS415-mCh-P4C | This study | MGTY096 |
| TB50A HIS3::pRS403-P(TEF1)-GFP-D4H | This study | MTY017 |
| TB50A HIS3::pRS403-P(TEF1)-mCherry-D4H | This study | MGTY104 |
| TB50a HIS3::pRS403-P(TEF1)-GFP-D4H URA3::pRS406-mCh-2xPH(PLCδ) | This study | MTY038 |
| TB50a URA3::GFP-D4H + pRS414-mCh-2xPH(PLCδ) | This study | BK4 |
| TB50α lam2::Nat lam4::Nat | This study | BK2-8 |
| TB50 lam2::Nat lam4::Nat HIS3::pRS403-P(TEF1)-GFP-D4H URA3::pRS406-mCh-2xPH(PLCδ) | This study | MGTY098 |
| TB50α lam2::Nat lam4::Nat URA3::GFP-D4H + pRS414-mCh-2xPH(PLCδ) | This study | MTY035 |
| TB50a LAM2-T518A LAM4-S401A | This study | PN001 |
| TB50A LAM2-T518A LAM4-S401A HIS3::P(TEF1)-GFP-D4H URA3::mCherry-2xPH | This study | MGTY102 |
| TB50α TOR1-1 AVO3-ΔCT::Hph HIS3::pRS403-P(TEF1)-GFP-D4H URA3::pRS406-mCh-2xPH(PLCδ) | This study | MGTY100 |
| TB50A URA3::mch-2xPH + pRS415-GFP-YSP2 | This study | MGTY107 |
| BY4741 P(VPS21)::URA3::P(SpNOP1)-GFP-VPS21 | PMID: 26928762 | SWAT_mCh-VPS21 |
| TB50 TRP1::YIPlac204TKC-DsRed-Express2-HDEL HIS3::pRS403-P(TEF1)-mCherry-D4H | This study | MGTY110 |
| TB50a hog1::Hph | Loewith lab | ΔHog1 |
| TB50α ysp2::NAT lam4::NAT hog1::Hph | This study | RL389-1b |
| TB50a ysp2::NAT lam4::NAT hog1::Hph | This study | RL389-1c |
| human cell lines | | |
| HBEC3-KT | PMID: 1560426; ATCC | CRL-4051 ™ |
| **Recombinant DNA** | | |
| All plasmids | | |
| pRS426-GFP-2xPH(PLCδ) | PMID: 1185441 | |
| pRS414-pTdh3-mCherry-2xPH(PLCδ) | This study | |
| pRS416-LactC2-GFP | PMID: 18187657 | |
| pRS415-mCh-P4C | Gift from Christopher Stefan lab | |
| pRS406-mCh-2xPH(PLCδ) | This study | |
| pRS403-P(TEF1)-GFP-D4H | This study | |
| pRS403-P(TEF1)-mCherry-D4H | This study | |

| Reagent/resource | Reference or source | Identifier or catalog number |
|---|---|---|
| pRS413-P(TEF1)-mCherry-D4H | This study | |
| pRS415-P(YSP2)-GFP-YSP2 | This study | |
| YIPlac204TKC-DsRed-Express2-HDEL | Addgene | #21770 |
| **Antibodies** | | |
| Sch9 (total) polyclonal rabbit | Agrobio - homemade | |
| Sch9 pS758 monoclonal mouse | Agrobio– homemade; Wanke et al, 2008 | |
| Ypk1 (total) polyclonal rabbit | Agrobio- homemade | |
| Ypk1 pT662 (1H3) monoclonal mouse | IGBMC; Berchtold et al, 2012 | |
| Akt1 (total) monoclonal mouse | Cell Signaling Technology | #2920 |
| Akt1 pS473 monoclonal rabbit | Cell Signaling Technology | #4060 |
| Donkey anti-rabbit (H + L) IRDye 680 (red) conjugated | Li-Cor Biosciences | 926-68073 |
| Donkey anti-rabbit (H + L) IRDye 800 (green) conjugated | Li-Cor Biosciences | 926-32213 |
| Donkey anti-mouse (H + L) IRDye 680 (red) conjugated | Li-Cor Biosciences | 926-68072 |
| Donkey anti-mouse (H + L) IRDye 800 (green) conjugated | Li-Cor Biosciences | 926-32212 |
| **Chemicals, enzymes, and other reagent drugs** | | |
| Rapamycin | LC laboratories or Thermo Fisher | R-5000 or J62473.MF |
| CK-666 | Sigma | SML0006-25MG |
| Atorvastatin | Sigma | PHR1422 |
| Fluconazole | Sigma | Y0000557 |
| **SAR screen substances** | | |
| Carnitine | Sigma | C0283 |
| Palmitate (C16) | Sigma | P0500 |
| Palmitoylglycine (C16-Glycine) | Sigma | 870817P |
| Palmitoylglycerol (C16-Glycerol) | Sigma | 75614 |
| Palmitoylcholine (C16-Choline) | Cayman | Cay9003456-5 |
| PAF | Enzo Life Sciences | BML-L100-0005 |
| LauroylCarnitine (C12-C) | Sigma | 39953 |
| MyristoylCarnitine (C14-C) | Sigma | 61367 |
| PalmitoylCarnitine (C16-C) | Sigma | P4509 |
| StearoylCarnitine (C18-C) | Sigma | 61229 |
| OleoylCarnitine (C18:1-C) | Sigma | 870852 P |
| L-*erythro*-DHS | Brunschwig | CAY24374 |
| D-*erythro*-DHS | Brunschwig | CAY10007945 |
| PHS | Sigma | P2795 |
| **Software** | | |

| Reagent/resource | Reference or source | Identifier or catalog number |
|---|---|---|
| FIJI 1.54 f | ImageJ | |
| Microsoft Excel | Microsoft | |
| Prism 10.2.3 | GraphPad | |
| MetaXpress software (Ver 6.7.0.211) | MetaXpress | |
| Cellpose3 | Stringer et al, 2021 Stringer and Pachitariu, 2025 | |
| Origin Pro 2024 | OriginLab Corp. | |
| **Other** | | |
| Bovine pituitary extract | Life Technologies | 17005042 |
| Halt Protease & Phosphatase Inhibitor Cocktail | Thermo Fisher Scientific | |
| iBlot2 Gel Transfer System | Thermo Fisher Scientific | |
| Odyssey imaging system | LI-COR | |
| Complete protease inhibitor cocktail | Roche Molecular Biochemicals | |
| VI 0.4 µ-Slides | Ibidi | |
| Paraformaldehyde 3% Glutaraldehyde 0.35% In 0.1 M Sodium Cacodylate Buffer pH 7.4 | Electron Microscopy Sciences | 15949-60 |
| Lipidspot™ 488 | Biotium | 70065 |
| CherryTemp temperature controller | Cherry Biotech | |

## Yeast cell culture and treatments

Yeast strains were generally generated using classical recombination techniques. Strains with point mutations were generated using CRISPR-Cas9-based methods.

Yeast cells were routinely grown at 30 °C in synthetic media (SC) buffered at pH 6.2 with 0.1 M Sorensen Buffer and supplemented with 2% glucose and appropriate amino acids and nucleobases to maintain plasmids. For microscopy experiments, cells were grown in low-fluorescence media. All experiments were performed in logarithmically growing cells. Solvents and stock concentrations for each treatment substance are listed in Appendix Table S1. For small amphipath treatments, cells were usually grown to $OD_{600\,nm}$ 0.6–0.8 and treated with the indicated substances at 5 µM, or an equivalent volume of vehicle (DMSO or Methanol). To test the impact of culture density on PalmC effect, cells were pelleted by centrifugation, resuspended to $OD_{600\,nm}$ 0.1 or 2 in fresh prewarmed media, and incubated for 15–20 min at 30 °C to recover before administering 5 µM PalmC. To negate the effect size variation caused by offsets in the small amphipath/cell ratio, all experiments with small amphipaths were performed at the same $OD_{600\,nm}$ for the compared conditions. For hyperosmotic shocks, cultures were diluted with prewarmed media containing 2 M sorbitol to a final concentration of 0.5 M or 1 M sorbitol. For CK666 treatment, cultures were treated with 250 µM CK-666 or an equivalent volume of vehicle (DMSO). For heat shock, cells were grown at 24/25 °C, 12 ml subcultures were

portioned for each timepoint, and transferred to a gyratory water bath preheated to 37 °C. For pharmacological ergosterol depletion, logarithmically growing cells were supplemented at low $OD_{600\,nm}$ with 100 μM Atorvastatin or Fluconazole or an equivalent volume of vehicle (DMSO). Samples were harvested at the indicated timepoints and processed as described below.

## Mammalian cell culture and treatments

HBEC3-KT (human bronchial epithelium immortalized with hTERT) cells were kindly provided by Prof. Georgia Konstantinidou. They were cultured in Keratinocyte serum-free medium (SFM) supplemented with human recombinant epidermal growth factor (rEGF) and bovine pituitary extract at 37 °C with 5% $CO_2$. Cells were authenticated by Microsynth and tested negative for mycoplasma by GATC Biotech. For WB experiments, mammalian cells were seeded on 6-well plates a day prior to the experiment. Treatments were performed by exchanging culture media for prewarmed media supplemented with the indicated substances. At the indicated timepoints, cells were rinsed with ice-cold PBS quickly before lysis (described below).

## WB sample preparation and detection of phosphoproteins on WB

Yeast culture aliquots were processed according to standard TCA-urea extraction procedures. In brief, culture aliquots were incubated with 5% TCA on ice for at least 10 min, before pelleting and drying cells with ice-cold acetone. Cells were lysed by bead beating in a urea buffer (25 mM Tris pH 6.8, 6 M Urea, 1% SDS), and boiled for 5 min. Denatured lysates were mixed with 2x sample buffer (125 mM TRIS pH 6.8, 20 vol% glycerol, 2% SDS, 0.02% Bromphenol Blue, 200 mM DTT), and boiled again before analyzing them via WB. Mammalian cells were lysed with a lysis buffer (40 mM HEPES, 10 mM Na-PPi, 10 mM Na-β- glycerophosphate, 4 mM EDTA 1% Triton X-100, pH 7.4) supplemented with 1× Halt Protease & Phosphatase Inhibitor Cocktail (Thermo Fisher Scientific) and kept at −20 °C until further analysis. A Pierce BCA Protein Assay (Thermo Fisher Scientific) was used to determine protein concentration in the samples. In all, 15–50 μg of total protein of each sample was used for further analysis. The 5x sample buffer (312.5 mM Tris, 10% SDS, 50% glycerol, 25% β-mercaptoethanol, 0.1% bromophenol blue, pH 6.8) was added, and samples were denatured at 95 °C for 5 min. Protein lysates were separated on 7.5% (Ypk1, Sch9) or 10% (Ypk1, Akt) SDS-page gels and blotted onto nitrocellulose membranes using the iBlot2 Gel Transfer System (Thermo Fisher Scientific). Membranes were blocked with BSA, incubated with primary antibodies overnight at 4 °C, washed, and incubated with secondary antibodies for 1 h at room temperature, using PBS- (yeast WBs) or TBS (yeast and mammalian WBs) -based buffers. Membranes were developed using an Odyssey imaging system (LI-COR), and signal intensities were quantified using FIJI (ImageJ 1.54 f). Calculations were performed in Microsoft Excel, and data were plotted using GraphPad Prism (10.2.3).

## Yeast membrane extraction

1 L culture was harvested by centrifugation (5 min, 5000 × g), and resuspended in 80 mL 0.4 M sucrose Buffer A - 25 mM imidazole (pH 7.0) containing 2.5 μg/mL pepstatin A (Sigma, St. Louis, MO),

and a 1/100 dilution of complete protease inhibitor cocktail (Roche Molecular Biochemicals, Basel, Switzerland) from a stock in $H_2O$ (two tablets dissolved in 2 mL $H_2O$). After centrifugation (5000 × g for 10 min), the pellet was covered by twice its volume of glass beads followed by just enough 0.4 M sucrose in buffer A to cover the cells and glass beads. The solution was vortexed three times for 2 min, then diluted three times in 0.4 M sucrose in buffer A and centrifuged at 530 × g for 20 min. The supernatant was recentrifuged at 20,000 × g for 30 min. The pellet from this last step was then resuspended in 2 mL buffer A, and 1 mL aliquots were loaded onto discontinuous sucrose gradients and centrifuged overnight (14 h) at 80,000 × g and 4 °C. Membranes banding at the 2.25/1.65 M sucrose interfaced were collected, diluted 4 times in buffer A and pelleted by centrifugation at 30,000 × g and 4 °C for 40 min. They were stored and −20 °C.

## Metabolite extraction

Frozen cell lysates and membrane fraction from yeast were pre-extracted and homogenized by the addition of 1 mL of MeOH:$H_2O$ (4:1), in the Cryolys Precellys 24 sample Homogenizer (2 × 20 s at 10,000 rpm, Bertin Technologies, Rockville, MD, USA) with ceramic beads. The bead beater was air-cooled down at a flow rate of 110 L/min at 6 bars. Homogenized extracts were centrifuged for 15 min at 4000 × g at 4 °C (Hermle, Gosheim, Germany). The supernatant (metabolite extract) was collected and evaporated to dryness in a vacuum concentrator (LabConco, Missouri, USA). Dry cell and plasma membrane extracts were reconstituted in 300 μL of MeOH:$H_2O$ (4:1). For absolute quantification of PalmC the samples were prepared by mixing an aliquot (5 μL) of reconstituted extract with 250 μL of the ice-cold internal standard solution (in 100% methanol) and 45 μL of 0.1% formic acid in water. These mixtures were centrifuged, and the supernatant was injected for LC-HRMS analysis in positive ionization mode.

## Protein quantification for normalization of PalmC levels

Following the metabolite extraction, the excess of organic solvent remaining on the top of precipitated protein pellets was evaporated and the protein pellets were resuspended in the lysis buffer containing: 20 mM Tris-HCl (pH 7.5), 4 M guanidine hydrochloride, 150 mM NaCl, 1 mM $Na_2EDTA$, 1 mM EGTA, 1% Triton, 2.5 mM sodium pyrophosphate, 1 mM beta-glycerophosphate, 1 mM $Na_3VO_4$, 1 μg/ml leupeptin; using brief probe-sonication (5 pulses x 5 s). BCA Protein Assay Kit (Thermo Scientific, MA, USA) was used to measure (A562nm) total protein concentration (Hidex, Turku, Finland).

## Hydrophilic interaction liquid chromatography coupled to high-resolution mass spectrometry (HILIC-HRMS)

For PalmC quantification, the prepared extracts were analyzed by Hydrophilic Interaction Liquid Chromatography coupled to high-resolution mass spectrometry (HILIC-HRMS) in positive ionization mode. A Vanquish Horizon (Thermo Fisher Scientific) ultra-high performance liquid chromatography (UHPLC) system coupled to Q Exactive™ Focus interfaced with a HESI source was used for the quantification of amino acids. Chromatographic separation was carried out using an Acquity BEH Amide (1.7 μm, 100 mm × 2.1 mm I.D.)

column (Waters, Massachusetts, US). The mobile phase was composed of A = 20 mM ammonium formate and 0.1% formic acid in water and B = 0.1% formic acid in ACN. The gradient elution started at 95% B (0–2 min), decreasing to 65% B (2 min–14 min), reaching 50% B at 16 min, followed by an isocratic step (16 min – 18 min) and 4 min post-run for column re-equilibration. The flow rate was 400 μL/min, column temperature 25 °C and the sample injection volume was 2 μl. HESI source conditions operating in positive mode were set as follows: sheath gas flow at 60, aux gas flow rate at 20, sweep gas flow rate at 2, spray voltage at +3 kV, capillary temperature at 300 °C, s-lens RF level at 60 and aux gas heater temperature at 300 °C. Full scan HRMS acquisition mode ($m/z$ 50–750) was used with the following MS acquisition parameters; resolution at 70,000 FWHM, 1 microscan, 1e6 AGC and 100 ms as maximum inject time. Data were processed using Xcalibur (version 4.1, Thermo Fischer Scientific). For absolute quantification, calibration curve and the stable isotope-labeled internal standard (ISTD–PalmC-d9) was used to determine the response factor. Linearity of the standard curves was evaluated using a 9-point range; in addition, peak area integration was manually curated and corrected where necessary. The concentration of PalmC was corrected for the ratio of peak intensity (peak area) between the analyte and the ISTD, to account for matrix effects. Measured metabolite quantities were normalized to protein content and expressed in nmoles/mg of protein, to normalize for sample amount differences.

## SAR screen live-cell fluorescent microscopy

For live-cell imaging, TB50A WT cells expressing GFP-2xPH$^{PLC\delta}$ from plasmid were mounted onto Concanavalin A-coated VI 0.4 μ-Slides (Ibidi) primed with media (except LCBs). For substance treatment, media were removed and exchanged with small amphipathic-containing media at RT. Images of the equatorial plane were taken at the indicated timepoints on a Leica TCS SP5 gated STED CW microscope with a 63.0 × 1.40 oil immersion objective and LAS AF (Ver. 2.7.3.9723) software. The effects of LCBs were extremely variable under these conditions, thus D-DHS and PHS were instead added to 5 μM to shaking yeast cultures, cells were concentrated by brief centrifugation at the indicated timepoints, mounted onto glass slides, and imaged using a Zeiss LSM 800 microscope with a 63 × 1.40 oil immersion objective and Zen Blue (Ver. 2.6) software. Images were processed in FIJI (ImageJ 1.54 f).

## SAR screen PM invagination quantification by automated confocal microscopy

The screen was performed in collaboration with the ACCESS facility in Geneva. TB50A WT cells expressing mCherry-2xPH$^{PLC\delta}$ from the HIS3 locus were grown logarithmically and set to OD$_{600\,nm}$ 0.6. The culture was split into subcultures of the same volume, agitating at 30 °C, and substances were added to 5 μM from prewarmed stocks. At indicated timepoints, culture aliquots were taken and mixed 1:2 with Paraformaldehyde 3% Glutaraldehyde 0.35% In 0.1 M Sodium Cacodylate Buffer pH 7.4 (Electron Microscopy Sciences). Cells were fixed for 30-60 min at RT and diluted in low-fluorescence media. C14-C invaginations could not be preserved in fixed cells. Images of the equatorial plane were taken on a Molecular Devices™ ImageXpress Micro HTai C® High-Content Imaging System equipped with a 60 × 1.25 NA water immersion objective. Single-cell and PM invagination segmentation and analysis was performed with custom module editor from MetaXpress software (Ver 6.7.0.211). Briefly, individual yeasts were segmented on a log filtered transmitted light image. The PM invaginations were segmented on the mCherry-2xPH image after top hat deconvolution. The area sum of PM invaginations was normalized to the cell area and averaged for each condition and timepoint. These average values were then normalized to the value determined in vehicle-treated cells at the same timepoint for each experiment. Calculations were performed in Microsoft Excel, and data were plotted using GraphPad Prism (10.2.3).

## Live-cell microscopy and treatments (except SAR)

For microscopy experiments, WT or mutant yeast cells were generally grown to mid-log phase in low-fluorescence media and imaged at room temperature (except Heat shock) on a Leica STELLARIS 8 FALCON FLIM Microscope with a 63 × 1.4 Oil immersion objective and LAS X (Ver. 4.6.1.27508) software (except Heat shock). If not stated otherwise, in preparation for imaging and treatments, cells were mounted onto Concanavalin A-coated VI 0.4 μ-Slides (Ibidi) primed with media and gently washed with media once to obtain a single layer of cells.

For FLARE microscopy, mCherry-2xPH$^{PLC\delta}$ + GFP-D4H were generally expressed from genomically integrated copies (except for heat shock, where mCherry-2xPH$^{PLC\delta}$ was expressed from plasmid). All other FLARE combinations were expressed from plasmids. For FM4-64 staining of PM and endosomes, cells were stained on slide briefly (30–60 s) with 10 μM FM4-64 before replacing the media and incubating the cells at RT for ~10 min until endosomes became visible. For Lipidspot™ 488 staining of lipid droplets, cells were stained on slide two consecutive times for 5 min with media supplemented 1:1000 with Lipidspot™ 488, and treatments were performed in Lipidspot™ 488-containing media.

For vehicle, PalmC, Rapamycin, 1 M sorbitol, and CK-666 treatment, several positions were selected proximal to the "+" channel opening, and the same positions/cells were imaged at t0, and at the indicated timepoints post treatment. For treatments, media supplemented with indicated substances were added to the "+" channel opening, while the excess was absorbed with a paper tissue inserted into the "−" channel end. To test the effects of C16-Glycerol, C16-Choline, and PHS on yeGFP-D4H localization, 5 μM of these substances were instead added to mid-log phase liquid cultures to avoid the previously observed substance absorption to the slide. Before the indicated timepoint, cells were concentrated by short centrifugation and mounted onto high-precision glass coverslips for imaging.

For heat shock, cells were grown at 24 °C, mounted onto Concanavalin A-coated 15H glass coverslips primed with media, washed gently with media to remove floating cells, and heat shock was performed using a CherryTemp temperature controller (Cherry Biotech), while recording equatorial plane videos on a Olympus IX83 widefield microscope equipped with a 100×/NA1.4 objective and an ImageEM X2 EM-CCD camera (Hamamatsu) under the control of the VisiView software (Visitron Systems). Excitation and emission were filtered using a 49008_Zeiss Axio_un-mounted ET/mCherry/TexasRed ET560/40x ET630/75 m T585lp

and 49002_Zeiss Axio_un-mounted ETGFP (FITC/CY2) ET470/40x ET525/50 m T495LPXR. As the temperature shift causes a change in optical conditions, visible as an overall decrease of fluorescence signal, the sample was shifted back to 24 °C after 10 min to record final state images with the original optical conditions.

FM4-64, dsRED-HDEL, Lipidspot (alone), and GFP-Ysp2 data were generally recorded as z-stacks and deconvolved with Lightning deconvolution (Leica). All other data were recorded as equatorial plane images. Images were processed using FIJI (ImageJ 1.54 f). Before the generation of kymographs, timelapses of selected cells were stabilized using the Image Stabilizer plugin for ImageJ.

### FLARE colocalization analysis

In flow chambers, we typically observed different and more variable small amphipath kinetics than in liquid culture. We decided to use the 5 min timepoint for quantifications with PalmC, as it generally displayed a robust phenotype. To calculate colocalization between $2xPH^{PLC\delta}$ and other FLAREs, cells were first segmented using Cellpose3 (Stringer et al, 2021; Stringer and Pachitariu, 2025) and then analyzed in FIJI (ImageJ 1.54 f). Cells that were intersected by image borders were excluded. Pearson's correlation coefficient (PCC) as a score for colocalization between different FLAREs and $2xPH^{PLC\delta}$ in single cells was calculated using the BIOP version of the JACoP plugin (Bolte and Cordelières, 2006). Data were analyzed and plotted using GraphPad Prism (10.2.3).

### FLARE sorting coefficient analysis

PM invagination sorting coefficients were calculated for each flare using $2xPH^{PLC\delta}$ as reference with the following equation:

$$Sorting\ coefficient = \frac{(F_{tested\ FLARE}/F_{2xPH^{PLC\delta}})\text{PM invagination}}{(F_{tested\ FLARE}/F_{2xPH^{PLC\delta}})\text{PM}}$$

where $F_{tested\ FLARE}$ and $F_{2xPH^{PLC\delta}}$ are the background-subtracted fluorescence densities integrated from the plot profiles of PM invaginations with unsaturated signal or an adjacent non-invaginated PM patch in the same cell. Only one invagination was measured in each cell. Microscopy images were visualized, and data extracted using FIJI (ImageJ 1.54 f). Data were analyzed using Origin Pro 2024 (OriginLab Corp.), and plots were generated using GraphPad Prism (10.2.3).

### GFP-D4H plasma membrane/cytoplasm ratio quantification

To evaluate plasma membrane enrichment in of GFP-D4H during heat shock, cells were analyzed in one frame before heat shock (t0, 24 °C), and in one frame after 10 min of heat shock (37 °C) and subsequent back-shift to 24 °C to have the same optical conditions. Single cells were first segmented using Cellpose3 (Stringer et al, 2021; Stringer and Pachitariu, 2025), before analysis was performed in FIJI (ImageJ 1.54 f) using a custom macro: cells that were intersected by image borders were excluded. The segmented single-cell masks were used to generate both a ring-shaped peripheral ROI marking the PM and an eroded internal ROI marking the

cytoplasm. After removing fluorescent background by subtracting a median filtered (100px kernel) duplicate image, mean signal intensities ($\bar{x}$) were measured for both ROIs. The relative D4H enrichment at the PM ($\bar{x}$ GFP-D4H$_{PM}$/ $\bar{x}$ GFP-D4H$_{Cytoplasm}$) was calculated for each cell, and data were analyzed and plotted using GraphPad Prism (10.2.3).

### Lipid Droplet volume quantification

Cells were first segmented using Cellpose3 (Stringer et al, 2021; Stringer and Pachitariu, 2025), before Lipid droplet volumes were measured from Lightning-deconvolved z-Stacks of Lipidspot™ 488-stained cells in FIJI (ImageJ 1.54 f) using the 3D Objects Counter plugin. The average total LD volume/cell ($\mu m^3$) was determined for each experiment and condition, and data were analyzed and plotted using GraphPad Prism (10.2.3).

### Statistics and reproducibility

Western blot results for TORC2 and TORC1 activity (pT662 Ypk1 and pS758 Sch9, respectively) are presented as mean values and SD from at least three independent replicates (or at least two replicates for supplementary figures). SAR live-cell microscopy experiments were performed three times for each compound with similar results, and representative cells are shown. The SAR high-content microscopy screen was performed three times, and the average, vehicle-normalized value for each condition is plotted together with the mean and SD of all three experiments. To assess differences in FLARE colocalization upon PalmC treatment in WT cells, the experiment was repeated three times, analyzed in single cells, and statistical significance was calculated from the mean Pearson's correlation coefficients using unpaired $t$ test after confirming normal distribution using Shapiro–Wilk test. Observations of D4H behavior under different conditions were repeated at least three times with similar results. Single-cell scatter plots represent pooled datasets from two to three experiments (color coded), to analyze large numbers of cells per condition (n counts are indicated in the graphs); plotted with median and interquartile range (bottom 25% to top 75% quartile). Sorting coefficient values were determined from one invagination/cell; the values from two to three independent experiments (color coded, $n$ counts indicated in the graphs) are plotted together with median (line), interquartile range (bottom 25% to top 75% quartile, box), and range (whiskers). Lipid droplet volumes were determined in three separate experiments for single cells, averaged for each experiment, and mean and SD of the three experiments were plotted. Supplementary microscopy experiments were performed at least twice with similar results.

## Data availability

This study includes no data deposited in external repositories.

The source data of this paper are collected in the following database record: biostudies:S-SCDT-10_1038-S44318-025-00618-7.

# Peer review information

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

## Acknowledgements

RL acknowledges support from the Canton of Geneva, the Swiss National Science Foundation (project 310030_207754), and the European Research Council (ERC AdG TENDO). AR acknowledges funding from the Swiss National Fund for Research grant numbers #CRSII5_189996 and #310030_200793 and the European Research Council Synergy grant number #951324-R2-TENSION. RL and AR further acknowledge support from the SNSF NCCR Chemical Biology. This work used the Photonics imaging platform, and the ACCESS platform at the University of Geneva. Metabolite extraction, protein quantification, and HILIC-HRMS for PalmC quantification were performed and evaluated at the Metabolomics platform at the University of Lausanne. We thank Margot Riggi for collecting the yeast cell lysates and membrane fractions for PalmC quantification, Kerstin Hinterndorfer for generating the pRS406-mCh-2xPH(PLCδ) plasmid and the original mCh-2xPH(PLCδ) yeast strain, Ariane Bergmann for generating CRISPR mutants, and Paraskevi Linardou for technical support in generating yeast strains. We thank Marko Kaksonen, Sophie Martin, Christopher Stefan, and Tim Levine labs for sharing plasmids and/or materials, and, together with Loewith and Roux labs, for helpful discussions.

## Author contributions

**Maria G Tettamanti**: Conceptualization; Data curation; Formal analysis; Validation; Investigation; Visualization; Methodology; Writing—original draft; Writing—review and editing. **Paulina Nowak**: Conceptualization; Formal analysis; Investigation; Visualization; Writing—review and editing. **Beata Kusmider**: Conceptualization; Formal analysis; Validation; Investigation; Methodology. **Jennifer M Kefauver**: Conceptualization; Supervision; Validation; Investigation; Methodology; Writing—review and editing. **Vincent Mercier**: Data curation; Formal analysis; Validation; Investigation; Visualization; Methodology; Writing—review and editing. **Aurélien Roux**: Conceptualization; Resources; Supervision; Funding acquisition; Writing—original draft; Project administration; Writing—review and editing. **Robbie Loewith**: Conceptualization; Resources; Supervision; Funding acquisition; Writing—original draft; Project administration; Writing—review and editing.

Source data underlying figure panels in this paper may have individual authorship assigned. Where available, figure panel/source data authorship is listed in the following database record: biostudies:S-SCDT-10_1038-S44318-025-00618-7.

## Disclosure and competing interests statement

The authors declare no competing interests.

# Expanded View Figures

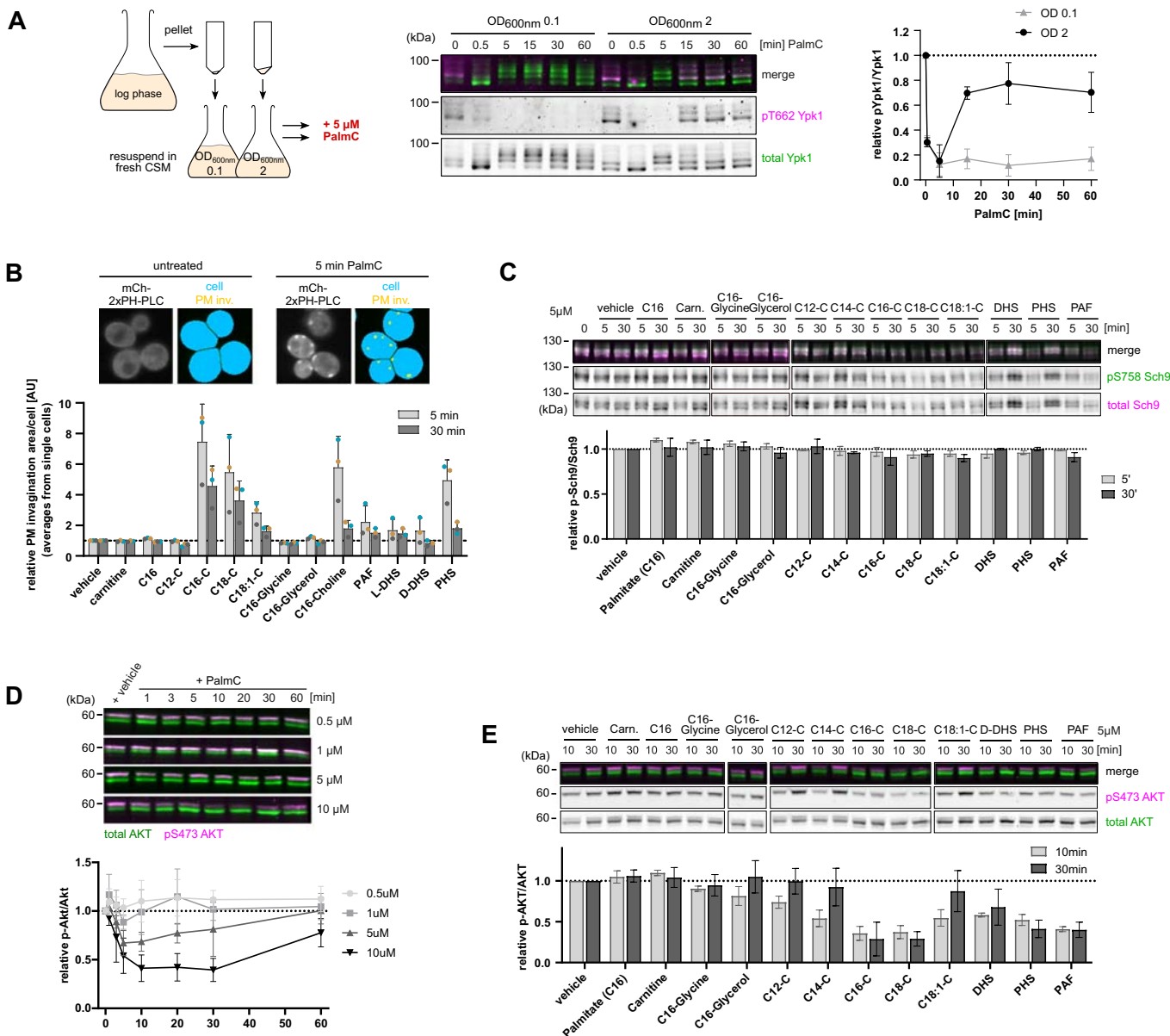

**Figure EV1. Additional characterization of the effects of small amphipaths in yeast and mammalian cells.**

(A) Western blot analysis of the effect of different yeast culture densities on PalmC-induced inhibition of TORC2. WT cells were pelleted and resuspended in fresh media to either OD 0.1 or OD 2 before treatment with 5 μM PalmC, and TORC2 activity was assessed by relative phosphorylation of Ypk1. Mean and SD of $N = 3$ independent experiments. (B) PM invagination screen in fixed WT cells expressing mCh-2xPH$^{PLC\delta}$. Cells were treated with the indicated substances at 5 μM, and samples were taken at the indicated timepoints and fixed. Representative images and segmentations of untreated cells and cells treated with PalmC for 5 min are shown. The plot shows the relative cell area fraction occupied by PM invaginations under different treatments. Determined values for each condition were normalized to the same vehicle timepoint for each experiment. Mean and SD of $N = 3$ independent experiments. (C) Western blot analysis showing the effect of different PalmC derivatives on TORC1 activity. WT cells were treated with indicated substances at 5 μM for 5 or 30 min, and TORC1 activity was assessed by relative phosphorylation of Sch9. Mean and SD of $N = 3$ independent experiments. (D, E) Western blot analysis of mTORC2 activity in HBEC3-KT cells. Cells were treated with (D) different concentrations of PalmC, or (E) with the indicated PalmC derivatives at 5 μM, and mTORC2 activity was assessed by relative phosphorylation of AKT. Mean and SD of $N = 3$ independent experiments. Source data are available online for this figure.

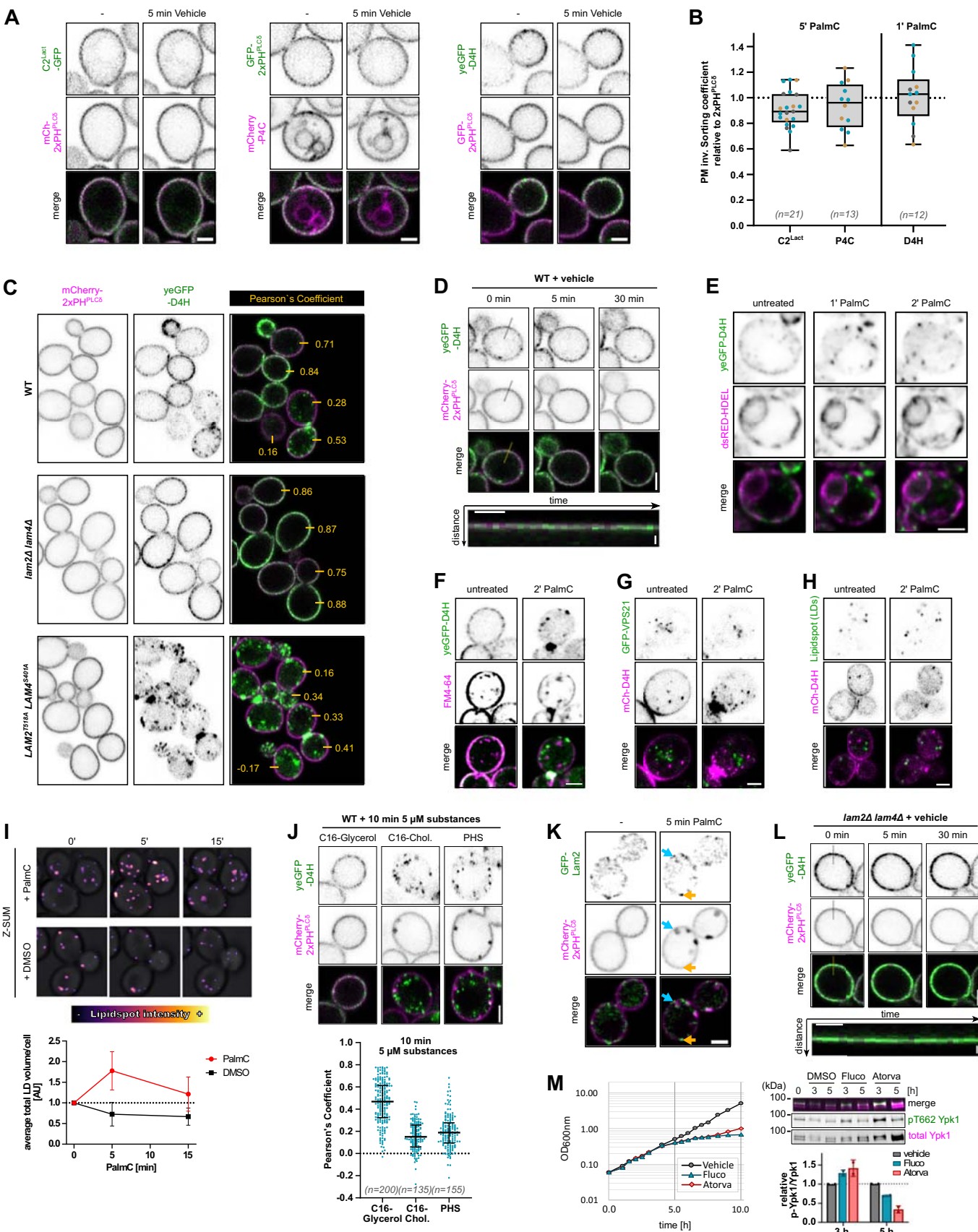

◄ **Figure EV2. Expanded observations with FLAREs on the effects of PalmC and sterol depletion in yeast cells.**

(A) Live-cell fluorescence microscopy of yeast cells expressing a PI(4,5)P2 reporter (GFP- or mCh-2xPH$^{PLC\delta}$) along with a phosphatidylserine (PS) reporter (C2$^{Lact}$-GFP), a PI4P reporter (mCh-P4C), or a free ergosterol reporter (yeGFP-D4H). Cells were treated with vehicle for 5 min. Scale bar = 2 μm. (B) Relative enrichment (sorting coefficients) of C2$^{Lact}$-GFP and mCh-P4C in PM invaginations after 5 min 5 μM PalmC treatment (left panel), or of yeGFP-D4H after 1 min 5 μM PalmC treatment (right panel), calculated using 2xPH$^{PLC\delta}$ as a reference. Single values from independent experiments (color coded) are plotted together with median, 25–75% interquartile range (box), and min-max range (whiskers). (C) Representative images showing yeGFP-D4H distribution and corresponding Pearson's Correlation Coefficients as measure of colocalisation with mCh-2xPH$^{PLC\delta}$ across the population in WT (top), *lam2Δ lam4Δ* (middle) or *LAM2$^{T518A}$ LAM4$^{S401A}$* (bottom) cells. (D) Live cell fluorescence microscopy of free ergosterol (yeGFP-D4H) and PI(4,5)P2 (mCh-2xPH$^{PLC\delta}$) in WT yeast cells before, and at indicated timepoints after the addition vehicle. Scale bar = 2 μm. The kymograph (bottom panel) depict yeGFP-D4H distribution relative to mCh-2xPH$^{PLC\delta}$-marked PM along the specified line over 30 min, at 1-min intervals. Scale bars: x = 5 min, y = 0.5 μm. (E–H) Live-cell fluorescence microscopy of yeast cells expressing a free ergosterol reporter (yeGFP- or mCherry-D4H), imaged before and 1–2 min after treatment with 5 μM PalmC. Cells are additionally also either (E) expressing dsRED-HDEL to mark ER (MAX projection of 3 equatorial slices, corresponding to a section of ~1.3 μm thickness), (F) stained with FM4-64 and incubated for 10 min to mark early endosomes (single equatorial slice) (G) overexpressing GFP-Vps21 to mark early endosomes (single equatorial slice), or (H) stained with Lipidspot™ 488 to mark lipid droplets (single equatorial slice). Scale bar = 2 μm. (I) Live-cell fluorescence microscopy of yeast cells stained with Lipidspot™ 488 to mark lipid droplets, and imaged before and after addition of 5 μM PalmC or vehicle at the indicated timepoints. Z-SUM projections of Lightning-deconvolved z stacks (merge brightfield and Lipidspot) are shown. Total lipid droplet volumes were determined in single cells, averaged, and the relative change to t0 was calculated. Mean and SD from N = 3 independent experiments. (J) Live cell fluorescence microscopy of free ergosterol (yeGFP-D4H) and PI(4,5)P2 (mCh-2xPH$^{PLC\delta}$) in yeast cells treated with indicated PalmC derivatives at 5 μM for 10 min. Scale bar = 2 μm. Scatter plots show the colocalization between the two probes, data points represent individual cells, plotted with median and 25%-75% interquartile range. Values from one representative experiment are shown. (K) Live cell fluorescence microscopy of yeast cells expressing a PI(4,5)P2 reporter (mCh-2xPH$^{PLC\delta}$) along with GFP-Lam2, before and after 5 μM PalmC treatment for 5 min. Scale bar = 2 μm. (L) Live cell fluorescence microscopy of free ergosterol (yeGFP-D4H) and PI(4,5)P2 (mCh-2xPH$^{PLC\delta}$) in *lam2Δ lam4Δ* yeast cells before, and at indicated timepoints after the addition vehicle. Scale bar = 2 μm. The kymograph (bottom panel) depict yeGFP-D4H distribution relative to mCh-2xPH$^{PLC\delta}$-marked PM along the specified line over 30 min, at 1-min intervals. Scale bars: x = 5 min, y = 0.5 μm. (M) Growth curve (left), and phospho-Ypk1 Western blot as readout for TORC2 activity (right), as measured simultaneously in logarithmically growing WT yeast cells treated either with 100 μM Atorvastatin, or 100 μM Fluconazole, or vehicle. The growth curve is a representative example from N = 1 experiment. The bar graph depicts relative Ypk1 phosphorylation; the values from N = 2 independent experiments are plotted with mean and SD. Source data are available online for this figure.

**A**

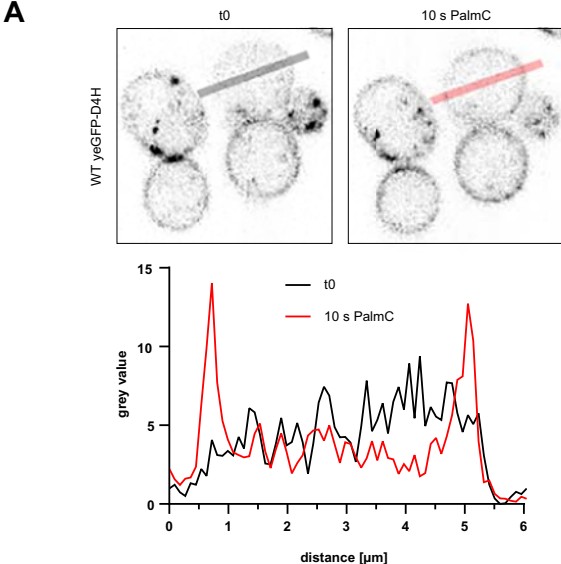

**B**

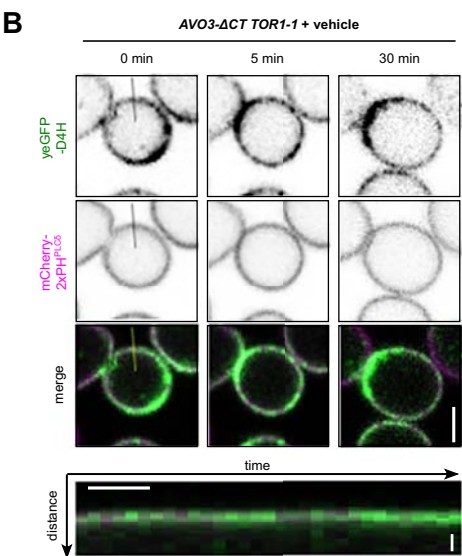

**C**

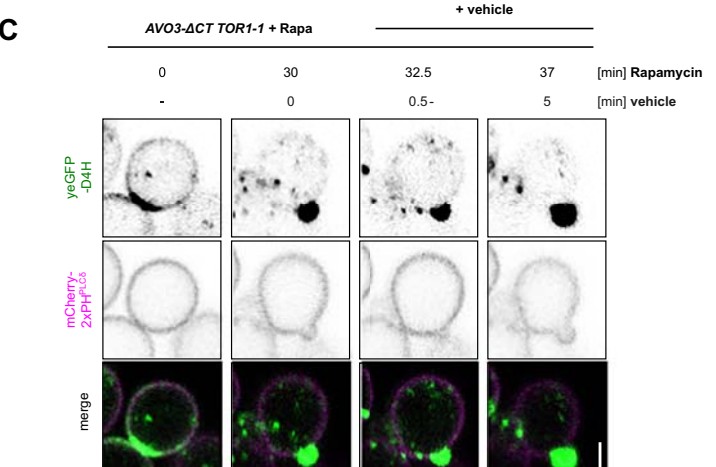

**D**

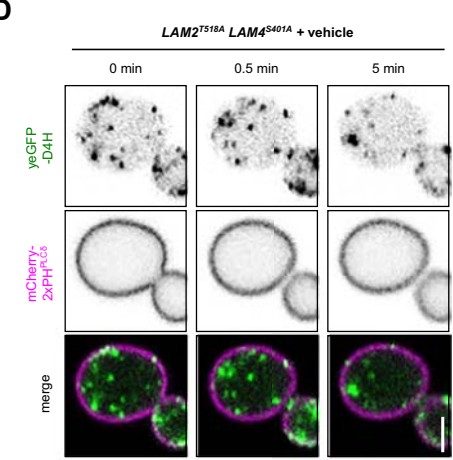

**Figure EV3. PalmC increases free PM ergosterol in WT cells, and vehicle control images for Fig. 3.**

(A) Live cell fluorescence microscopy of free ergosterol (yeGFP-D4H) in WT cells before and 10 s after 5 µM PalmC addition to WT cells. The line plot represents the average signal intensity of yeGFP-D4H along the specified line. (B–D) Live cell fluorescence microscopy of free ergosterol (yeGFP-D4H) and PI(4,5)P2 (mCh-2xPH$^{PLC\delta}$) (B) in *AVO3-ΔCT TOR1-1* cells before and after addition of 200 nm Rapamycin or (C) before and after 30 min pretreatment with 200 nM Rapamycin, followed by addition of vehicle, or (D) in *LAM2$^{T518A}$ LAM4$^{S401A}$* cells before and after addition of vehicle. Scale bar = 2 µm. The kymograph (bottom panel, if present) depicts yeGFP-D4H distribution relative to mCh-2xPH$^{PLC\delta}$-marked PM along the specified line over 30 min, at 1-min intervals. Scale bars: x = 5 min, y = 0.5 µm. Source data are available online for this figure.

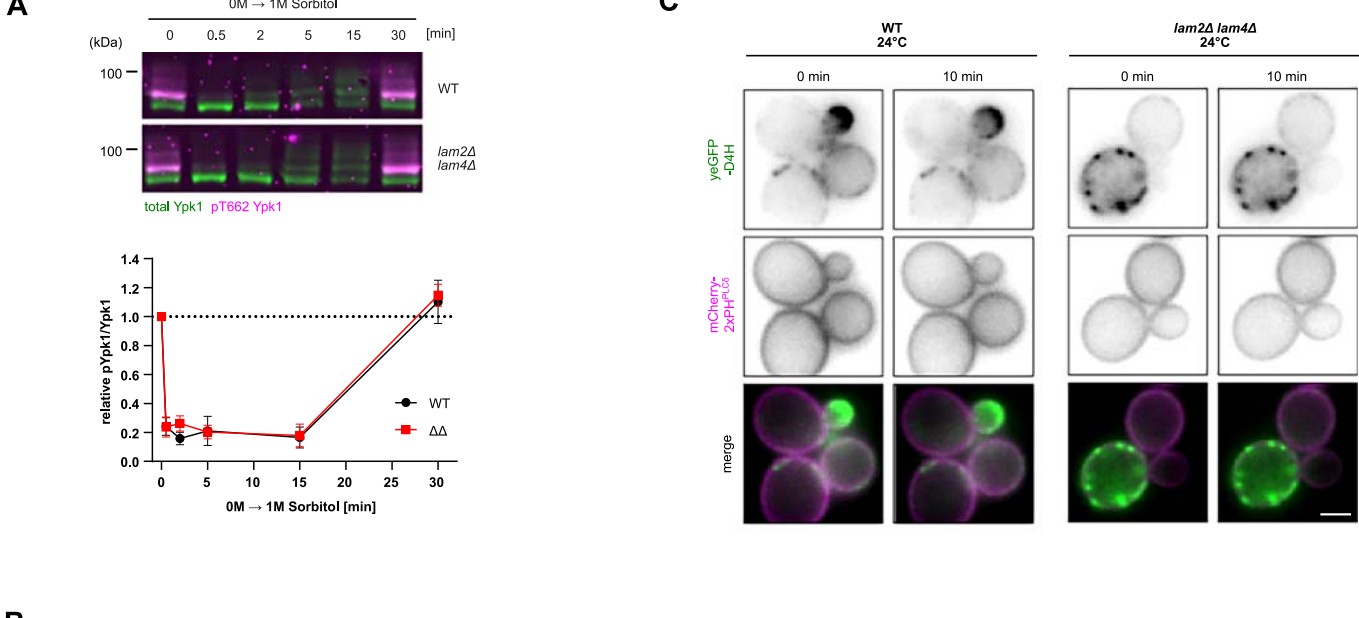

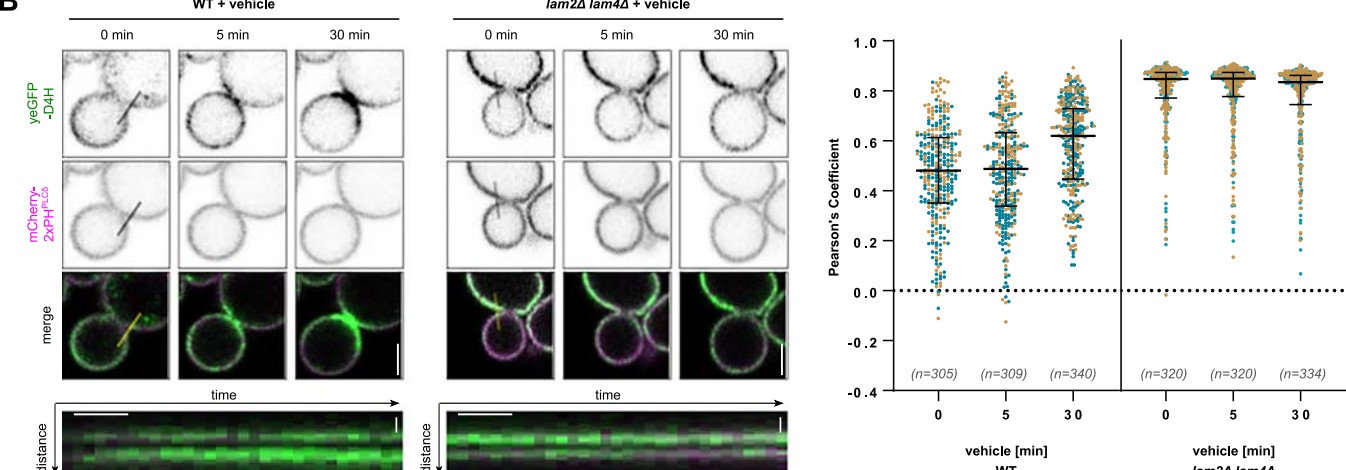

**Figure EV4.   TORC2 activity during severe hyperosmotic shock, and mock/vehicle controls for Fig. 4.**

(A) Western blot analysis of TORC2 activity in yeast cells. WT and *lam2Δ lam4Δ* cells were treated with 1 M Sorbitol and TORC2 activity was assessed by relative phosphorylation of Ypk1. Mean and SD of N = 3 independent experiments. (B, C) Live cell fluorescence microscopy of free ergosterol (yeGFP-D4H) and PI(4,5)P2 (mCh-2xPH$^{PLCδ}$) in (B) WT and *lam2Δ lam4Δ* cells before and after addition of vehicle (DMSO). Scale bar = 2 μm. The kymograph (bottom panel) depict yeGFP-D4H distribution relative to mCh-2xPH$^{PLCδ}$-marked PM along the specified line over 30 min, at 1-min intervals. Scale bars: x = 5 min, y = 0.5 μm. Scatter plots (if present) show the colocalization between the two probes before and at the indicated timepoints post-treatment. Data points represent individual cells, plotted with median and interquartile range. Different colors represent data from independent experiments. (C) WT and *lam2Δ lam4Δ* cells before and after 10 min of imaging at the idle growth temperature of 24 °C. Source data are available online for this figure.

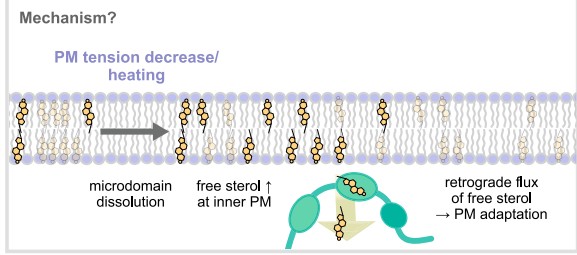

**Figure EV5. Speculative mechanism for PM stress recovery by sterol transport.**

Hyperosmotic shock and branched actin/endocytosis inhibition by CK-666 could both cause a dissolution of PM microdomains via a loss of PM tension. This would increase free PM sterols, which in combination with TORC2 inhibition could lead to an increased sterol retrograde transport through Lam2/4. Heat shock on the other hand could increase free PM sterols by thermic microdomain disruption.

