## [Peer Review File · The EMBO Journal]

A dynamic feedback loop between retrograde sterol transport and TORC2 controls adaptation of the plasma membrane to stress

Maria Tettamanti, Paulina Nowak, Beata Kusmider, Jennifer Kefauver, Vincent Mercier, Aurélien Roux, and Robbie Loewith

Corresponding author(s): Aurélien Roux (Aurelien.Roux@unige.ch) , Robbie Loewith (robbie.loewith@unige.ch)

Review Timeline:

Transfer from Review Commons:	8th Jul 25
Editorial Decision:	11th Aug 25
Revision Received:	2nd Sep 25
Accepted:	6th Oct 25

Editor: William Teale

Review
COMMONS

Transaction Report: This manuscript was transferred to The EMBO JOURNAL following peer review at Review Commons.

Review #1

1. Evidence, reproducibility and clarity:

Evidence, reproducibility and clarity (Required)

This is a very well conceived study of responses to plasma membrane stresses in yeast that signal through the conserved TORC2 complex.

Physical stress through small molecular intercalators in the plasma membrane is shown to be independent of their biochemistry and then studies for its effect on plasma membrane morphology and the distribution of free ergosterol (the yeast equivalent of cholesterol), with free being the pool of cholesterol that is available to probes and/or sterol transfer proteins. Experiments nicely demonstrate a negative feedback loop consisting of: stress -> increased free sterol and TORC2 inhibition -> activation of LAM proteins (as demonstrated by Relents and co-workers previously) -> removal of free sterol -> return to unstressed state of PM and TORC2.

Comments

Fig 2A: Is detection of PIP/PIP2/PS linear for target, or possibly just showing availability that is increased due to local positive curvature?

Can any marker be identified for the D4H spots at 2 minutes? In particular, are they early endosomes (shown by brief pre-incubation with FM4-64)?

Is there any functional (& direct) link between Arp inhibition (as in the Pombe study of LAMs by the lab of Sophie Martin) and PM disturbance by amphipathic molecules ?

Minor

Fig 2A: Labels not clear. Say for each part what FP is used for pip2.

Move fig s2d to main ms. The 1 min and 2 min data are integral to the story

The role of Lam2 and Lam4 in retrograde sterol transport has in vivo only been linked to one of their two StART domains not both, as mentioned in the text.

Throughout, images of tagged D4H should be labelled as such, not as "Ergosterol".

2. Significance:

Significance (Required)

These results in budding yeast are likely to be directly applicable to a wide range of eukaryotic cells, if not all of them. I expect this paper to be a significant guide for research in this area.

The paper specifically points out that the current experiments do not distinguish the precise causation among the two outcomes of stress: increased free sterol and TORC2 inhibition. Of these two outcomes which causes which is not yet known. If data were added that shed light on this causation that would make this work much more significant, but I can understand 100% that this extra step lies beyond - for a later study for which the current one forms the bedrock.

3. How much time do you estimate the authors will need to complete the suggested revisions:

Estimated time to Complete Revisions (Required)

(Decision Recommendation)

Less than 1 month

Yes

Review #2

1. Evidence, reproducibility and clarity:

Evidence, reproducibility and clarity (Required)

This manuscript describes multiple effects of positively-charged membrane-intercalating amphipaths (palmitoylcarnitine, PalmC, in particular) on TORC2 in yeast plasma

membranes. It is a "next step" in the Loewith laboratory's characterization of the effect of this agent on this system. The study confirms the findings of Riggi et al.(2018) that PalmC inhibits TORC2 and drives the formation of membrane invaginations that contain phosphatidylinositol-bis-phosphate (PIP2) and other anionic phospholipids. It also demonstrates that PalmC intercalates into the membrane, acts directly (rather than through secondary metabolism) and is representative of a class of cationic amphipaths. The interesting finding here is that PalmC causes a rapid initial increase in the plasma membrane ergosterol accessible to the DH4 sterol probe followed by a decrease caused by its transfer to the cytoplasm through its transporter, LAM2/4. TORC2 is implicated in these processes.

Loewith et al. have pioneered in this area and this study clearly shows their expertise. Several of the findings reported here are novel. However, I am concerned that PalmC may not be revealing the physiology of the system but rather adding tangential complexity. (This concern applies to the precursor studies using PalmC to probe the TORC2 system.) In particular, I am not confident that the data justify the authors' conclusions "...that TORC2 acts in a feedback loop to control active sterol levels at the PM and [the results] introduce sterols as possible TORC2 signalling modulators."

****Major issues****

1. The invaginations induced by PalmC may not be physiologic but simply the result of the well-known "bilayer couple" bending of the bilayer due to the accumulation of cationic amphipaths in the inner leaflet of the plasma membrane bilayer which is rich in anionic phospholipids. Such unphysiological effects make the observed correlation of invagination with TORC2 inhibition etc. hard to interpret.

2. Electrostatic/hydrophobic association of PIP2 with PalmC could sequester the anionic phospholipid(s). Such associations could also drive the accumulation of PIP2 in the invaginations. This could explain PalmC inhibition of TORC2 through a simple physical rather than biological process. So, it is difficult to draw any physiological conclusion about PIP2 from these experiments.

3. As the authors point out, a large number of intercalated amphipaths displace sterols from their association with bilayer phospholipids. This unphysiologic mechanism can explain how PalmC causes the transient increase in the availability of plasma membrane ergosterol to the D4H probe and its subsequent removal from the plasma membrane via LAM2/4. TORC2 regulation may not be involved. In fact, the authors say that "TORC2 inhibition, and thereby Lam2/4 activation, cannot be the only trigger for PalmC induced sterol removal." Furthermore, the subsequent recovery of plasma membrane ergosterol

could simply reflect homeostatic responses independent of the components studied here.

3a. The data suggest that LAM2/4 mediates the return of cytoplasmic ergosterol to the plasma membrane. To my knowledge, this is a nice finding that not been reported previously and is worth confirming more directly.

4. I agree with the authors that "It is unclear if the excess of free sterols itself is part of the inhibitory signal to TORC2..." Instead, the inhibition of TORC2 by PalmC may simply result from its artifactual aggregation of the anionic phospholipids (especially, PIP2) needed for TORC2 activity. This would not be biologically meaningful. If the authors wish to show that accessible ergosterol inhibits TORC2 activity or vice versa, they should use more direct methods. For example, neutral amphipaths that do not cause the aforementioned PalmC perturbations should still increase plasma membrane ergosterol and send it through LAM2/4 to the ER.

5. The mechanistic relationship between TORC2 activity and ergosterol suggested in the title, abstract and discussion is not secure. I agree with the concluding section of the manuscript called "Limitations of the study". It highlights the need for a better approach to the interplay between TORC2 and ergosterol.

****Minor issue****

Based on earlier work using the reporter fliptR, the authors claim that PalmC reduces membrane tension. They should consider that this intercalated dye senses many variables including membrane tension but also lipid packing. I suspect that, by intercalating into and thereby altering the bilayer, PalmC is affecting the latter rather than the former.

****Referees cross-commenting****

Reviewers #1 and #3 were much more impressed by this study than I was. I am not a yeast expert and so I may have missed or confused something. I would therefore welcome their expert feedback regarding my comments (#2). Ted Steck

2. Significance:

Significance (Required)

This is an interesting topic. However, use of the exogenous probe, palmitoylcarnitine, could be causing multiple changes that complicate the interpretation of the data.

3. How much time do you estimate the authors will need to complete the suggested revisions:

Estimated time to Complete Revisions (Required)

(Decision Recommendation)

Cannot tell / Not applicable

4. Review Commons values the work of reviewers and encourages them to get credit for their work. Select 'Yes' below to register your reviewing activity at Web of Science Reviewer Recognition Service (formerly Publons); note that the content of your review will not be visible on Web of Science.

Yes

Review #3

1. Evidence, reproducibility and clarity:

Evidence, reproducibility and clarity (Required)

The authors describe the effects of surfactant-like molecules on the plasma membrane (PM) and its associated TORC2 complex. Addition of the surfactants with a positively-charged headgroup and a hydro-carbon tail of at least 16 caused the rapid clustering of PI-4,5P2 together with PI-4P and phosphatidylserine in large membrane invaginations. The authors convincingly demonstrate that this effect of the surfactants on the PM is likely caused by a direct disturbance of the PM organization and/or lipid composition. Interestingly, upon PalmC treatment, free ergosterol of the PM was found to first concentrate in the clusters, but within <5min this ergosterol seemed to be transported into intracellular structures, causing an overall loss in free ergosterol of the PM. The authors speculate that the initial spike in free ergosterol might be the trigger for the shutdown of TORC2 signaling. The PalmC-triggered transport of free ergosterol from the PM to intracellular structures required the lipid transport proteins Lam2/4. Loss of these transporters caused a delay in TORC2 reactivation, supporting the idea that ergosterol transport out of the PM plays a role in the recovery of normal PM organization. Hyperosmotic shock mimics some of the effects observed with PalmC, but unlike PalmC treatment, TORC2 recovery after hyperosmotic shock is not dependent on Lam2/4.

The presented data are of high quality and most conclusions are well supported. However,

based on the presented data the model that a PalmC-triggered increase in free ergosterol is the cause of TORC2 inactivation is not obvious to me. The kinetics of the changes in free ergosterol levels and the changes in TORC2 activity do not match. Ergosterol is rapidly depleted after PalmC treatment (<5min) whereas TORC2 activity requires 30min to recover. Also, the hyperosmotic data on free ergosterol levels and TORC2 activity do not match. In fact, the presence of the large PM invaginations is a better predictor of TORC2 activity. The Lam2/4 data support the idea that ergosterol transport plays a role in the TORC2 recovery, but what role this is, is not clear to me. I think the data fit better with a model in which PalmC causes low tension of the PM which in turn disrupts normal lipid organization and thus causes TORC2 to shut down, maybe not by changes in free ergosterol but by changes, for instance, in lipid raft formation (which is in part effected by ergosterol levels). The transport of ergosterol is only one mechanism that is involved in restoring PM tension and TORC2 activity. However, sensing free ergosterol alone is most likely not the mechanism explaining how TORC2 senses PM tension. Therefore, I recommend that the model is revised (or supported by more data), reflecting the fact that free ergosterol levels do not directly correlate with the TORC2 activity, but instead might be only one of the PM parameters that regulate TORC2.

****Further comments:****

- If TORC2 is indeed inhibited by free ergosterol, the addition of ergosterol to the growth medium should be able to trigger similar effects as PalmC. If this detection of free ergosterol is very specific (e.g. if TORC2 has a binding pocket for ergosterol) we would expect that addition of other sterols such a cholesterol or ergosterol precursors should not inhibit TORC2.
- The experiment in Figure 1C is not controlled for differences in membrane intercalation of the different compounds. For instance, does C16 choline and C16 glycine accumulate at the same rate in the PM (measure similar to experiment in Figure 1B). Maybe the positive charge at the headgroup of the surfactants increases the local concentration at the PM and therefore can explain the difference in effect on the PM.
- Are the intracellular ergosterol structures associated (or in close proximity) with lipid droplets (ergosterol being modified and delivered into a lipid droplet)?
- How does the AA and DD mutations in Lam2/4 change the localization of the ergosterol sensor (before and after PalmC treatment).
- Does Lam2/4 localize to ER-PM contact sites near the large PM invaginations, which could allow for efficient transport of the free ergosterol that accumulates in these structures.

2. Significance:

Significance (Required)

The manuscript describes the effects of small molecule surfactants on the PM organization and on TORC2 activity. This is an important set of observation that helps understanding the response of cells to environmental stressors that affect the PM. This field of study is very challenging because of the limited tools available to directly observe lipids and their movements. I consider the data and most of its interpretations of high importance, but I am not convinced of the larger model that tries to link the ergosterol data with TORC2 activity. With adjustments of the model or additional experimental support, this manuscript will be of general interest for cell biologists, especially for researchers studying membrane stress response pathways.

3. How much time do you estimate the authors will need to complete the suggested revisions:

Estimated time to Complete Revisions (Required)

(Decision Recommendation)

Between 1 and 3 months

No

Reviewer #1 (Evidence, Reproducibility, and Clarity)

Reviewer comment:

This is a very well conceived study of responses to plasma membrane stresses in yeast that signal through the conserved TORC2 complex.

Physical stress through small molecular intercalators in the plasma membrane is shown to be independent of their biochemistry and then studies for its effect on plasma membrane morphology and the distribution of free ergosterol (the yeast equivalent of cholesterol), with free being the pool of cholesterol that is available to probes and/or sterol transfer proteins. Experiments nicely demonstrate a negative feedback loop consisting of: stress -> increased free sterol and TORC2 inhibition -> activation of LAM proteins (as demonstrated by Relents and co-workers previously) -> removal of free sterol -> return to unstressed state of PM and TORC2.

Author response:

We thank the reviewer for their positive and encouraging feedback. We are pleased to submit our revised manuscript and have addressed all points raised below.

Comment:

Fig 2A: Is detection of PIP/PIP2/PS linear for target, or possibly just showing availability that is increased due to local positive curvature?

Response:

This is an excellent and fundamental question. While FLARE signal likely reflects lipid availability, its detection is indeed influenced by factors such as membrane curvature and lipid composition, due to varying insertion depths of the lipid-binding domains. For example, studies using NMR suggest that the PLC δ PH domain partially inserts into membranes, potentially conferring curvature sensitivity (Flesch et al., 2005; Uekama et al., 2009). Similarly, curvature influences lactadherin binding, though it's unclear if this extends to its isolated C2 domain (Otzen et al., 2012; Shao et al., 2008; Shi et al., 2004). We could not find direct evidence for curvature sensitivity of P4C(SidC), but assume some influence exists. To avoid overinterpreting these limitations, we now describe our data based solely on the FLAREs used, rather than inferring enrichment of specific lipid species. We refer to these PM structures as "PI(4,5)P₂-containing", consistent with prior literature (Riggi et al., 2018) and have revised our manuscript accordingly.

Comment:

Can any marker be identified for the D4H spots at 2 minutes? In particular, are they early endosomes (shown by brief pre-incubation with FM4-64)?

Response:

We appreciate the reviewer's suggestion and have now added new data (Fig. S2E–H). We tested colocalization of D4H spots with FM4-64 (early endosomes), GFP-VPS21 (early endosome marker), and LipidSpot™ 488 (lipid droplets), but found no overlap. This later observation was not unexpected given that D4H does not recognize Sterol esters. D4H foci also did not overlap with ER (dsRED-HDEL), though they were frequently adjacent to it. While their exact identity remains unknown, we agree this is an intriguing direction for future investigation.

Comment:

Is there any functional (& direct) link between Arp inhibition (as in the Pombe study of LAMs by the lab of Sophie Martin) and PM disturbance by amphipathic molecules?

Response:

We have explored this connection and now present new data (see final paragraph of Results). Briefly, we show that CK-666 induces internalization of PM sterols in a Lam2/4-dependent manner, and that TORC2 activity is more strongly reduced in lam2 Δ lam4 Δ cells compared to WT. These findings support the idea that, like PalmC, Arp2/3 inhibition triggers a PM stress that is counteracted by sterol internalization.

Minor Comment:

Fig 2A: Labels not clear. Say for each part what FP is used for pip2.

Response:

As noted above, we revised image labels to clarify which FLAREs were used, and refer to data accordingly throughout.

Minor Comment:

Move fig s2d to main ms. The 1 min and 2 min data are integral to the story.

Response:

We agree and have incorporated the 1-min and 2-min data into the main figures. Vehicle-treated controls were moved to Fig. S2.

Minor Comment:

The role of Lam2 and Lam4 in retrograde sterol transport has in vivo only been linked to one of their two StART domains not both, as mentioned in the text.

Response:

Thank you for pointing this out. We have corrected the text to:

“[...]Lam2 and Lam4[...] contain two START domains, of which at least one has been demonstrated to facilitate sterol transport between membranes (Gatta et al., 2015; Jentsch et al., 2018; Tong et al., 2018).”

Minor Comment:

Throughout, images of tagged D4H should be labelled as such, not as "Ergosterol".

Response:

We have updated all relevant figure labels and text to refer to “D4H” rather than “Ergosterol”, in line with this recommendation.

Reviewer #1 (Significance):

These results in budding yeast are likely to be directly applicable to a wide range of eukaryotic cells, if not all of them. I expect this paper to be a significant guide of research in this area.

The paper specifically points out that the current experiments do not distinguish the precise causation among the two outcomes of stress: increased free sterol and TORC2 inhibition. Of these two outcomes which causes which is not yet known. If data were added that shed light on this causation that would make this work much more significant, but I can understand 100% that this extra step lies beyond - for a later study for which the current one forms the bedrock.

Response:

We thank the reviewer for their generous assessment. We agree that understanding the causality between increased free sterol and TORC2 inhibition is a critical next step.

Based on our current data, we believe the increase in free ergosterol precedes TORC2 inhibition. For example, TORC2 inhibition alone (e.g., via pharmacological means) does not initially increase free sterol, while it does enhance Lam2/4 activity, promoting sterol internalization (Fig. 3A). Baseline TORC2 activity also inversely correlates with free PM sterol levels in *lam2Δ lam4Δ* versus *LAM2^{T518A} LAM4^{S401A}* cells (Figs. 2D, S2C).

Additionally, during sterol depletion, we observe an initial increase in TORC2 activity before growth inhibition occurs, after which activity declines—likely due to compromised PM integrity (Fig. S2M). We now also show that adaptation to several other stresses (e.g., osmotic shock, heat shock, CK-666) partially depends on sterol internalization, which correlates with TORC2 activation (Fig. 4, S4B).

While these findings strengthen the model that PM stress perturbs sterol availability and secondarily impacts TORC2, we cannot yet definitively demonstrate causality. As suggested by Reviewer 3, we tested cholesterol-producing yeast (Souza et al., 2011), but found their response to PalmC indistinguishable from WT, making it difficult to draw mechanistic conclusions (Rebuttal Fig. 2).

Taken together, we favour a model where sterols affect PM properties sensed by TORC2, probably lipid-packing, rather than acting as direct effectors. We hope our revised manuscript more clearly conveys this model and serves as a strong foundation for future mechanistic studies.

Reviewer #2 (Evidence, Reproducibility, and Clarity)

Reviewer comment:

This manuscript describes multiple effects of positively-charged membrane-intercalating amphipaths (palmitoylcarnitine, PalmC, in particular) on TORC2 in yeast plasma membranes. It is a "next step" in the Loewith laboratory's characterization of the effect of this agent on this system. The study confirms the findings of Riggi et al.(2018) that PalmC inhibits TORC2 and drives the formation of membrane invaginations that contain phosphatidylinositol-bis-phosphate (PIP2) and other anionic phospholipids. It also demonstrates that PalmC intercalates into the membrane, acts directly (rather than through secondary metabolism) and is representative of a class of cationic amphipaths. The interesting finding here is that PalmC causes a rapid initial increase in the plasma membrane ergosterol accessible to the DH4 sterol probe followed by a decrease caused by its transfer to the cytoplasm through its transporter, LAM2/4. TORC2 is implicated in these processes. Loewith et al. have pioneered in this area and this study clearly shows their expertise. Several of the findings reported here are novel. However, I am concerned that PalmC may not be revealing the physiology of the system but rather adding tangential complexity. (This concern applies to the precursor studies using PalmC to probe the TORC2 system.) In particular, I am not confident that the data justify the authors' conclusions "...that TORC2 acts in a feedback loop to control active sterol levels at the PM and [the results] introduce sterols as possible TORC2 signalling modulators."

Author response:

We thank Reviewer #2 for the constructive and critical evaluation of our work. We appreciate the acknowledgment of the novelty and technical strength of several of our findings, and we understand the concern that PalmC could be eliciting non-physiological effects. Our study was designed precisely to use PalmC and similar membrane-active amphipaths as tools to strongly perturb the plasma membrane (PM) in a controlled and tractable way. We now state this intention explicitly in both the Introduction and Discussion sections. To address concerns about the specificity and physiological relevance of PalmC, we have expanded our dataset to include additional PM stressors (hyperosmotic shock, Arp2/3 inhibition, and heat shock), all of which reproduce key features observed with PalmC—namely, TORC2 inhibition, PM invaginations, and retrograde sterol transport (Fig. 4, S4). We hope this more comprehensive dataset, along with revised discussion and clarified claims, addresses the reviewer's concerns regarding physiological interpretation and artifact.

Major issues 1 and 2:

- 1. The invaginations induced by PalmC may not be physiologic but simply the result of the well-known "bilayer couple" bending of the bilayer due to the accumulation of cationic amphipaths in the inner leaflet of the plasma membrane bilayer which is rich in anionic phospholipids. Such unphysiological effects make the observed correlation of invagination with TORC2 inhibition etc. hard to interpret.*
- 2. Electrostatic/hydrophobic association of PIP2 with PalmC could sequester the anionic phospholipid(s). Such associations could also drive the accumulation of PIP2 in the invaginations. This could explain PalmC inhibition of TORC2 through a simple physical rather than biological process. So, it is difficult to draw any physiological conclusion about PIP2 from these experiments.*

Response to major issues 1 and 2:

We agree that amphipath-induced bilayer stress, including via the bilayer-couple mechanism, may contribute to PM curvature changes. However, the reviewer's assumption that PalmC inserts preferentially into the inner leaflet appears inconsistent with both literature and our observations. PalmC is zwitterionic, not cationic, and is unlikely to electrostatically sequester anionic lipids such as PIP2. For clarification, we included a short summary of our proposed mechanism of PalmC in the context of the current literature in our Discussion:

“[...] study it was also demonstrated that addition of phospholipids to the outer PM leaflet causes an excess of free sterol at the inner PM leaflet, and its subsequent retrograde transport to lipid droplets (Doktorova *et al.*, 2025). Although we cannot exclude that it is the substrate of a flippase or scramblase, PalmC is not a metabolite found in yeast, nor, given its charged headgroup, is it likely to spontaneously flip to the inner leaflet (Goñi, Requero and Alonso, 1996). Thus, we propose that PalmC accumulates in the outer leaflet, disrupts the lipid balance with the inner leaflet which is, similarly to the mammalian cell model (Doktorova *et al.*, 2025), rectified by sterol mobilization, flipping and internalization (Fig. 5B).”

While we agree that PM invaginations per se are not the central focus of this study, they are indeed a reproducible and biologically intriguing phenomenon. We emphasize that similar invaginations occur not only during PalmC treatment but also in response to other physiological stresses, such as hyperosmotic shock and Arp2/3 inhibition (Fig. 4), and have been reported independently by others (Phan *et al.*, 2025). Furthermore, related structures have been documented in yeast mutants with altered PIP2 metabolism or TORC2 hyperactivity (Rodríguez-Escudero *et al.*, 2018; Sakata *et al.*, 2022; Stefan *et al.*, 2002), and even in mammalian neurons with SJ1 phosphatase mutations (Stefan *et al.*, 2002). These observations support our interpretation that the observed invaginations represent an exaggerated manifestation of a physiologically relevant stress-adaptive process. In our previous study we indeed proposed that PI(4,5)P2 enrichment in PM invaginations was important for PalmC-induced TORC2 inactivation, using the heat sensitive PI(4,5)P2 kinase allele *mss4^{ts}* - a rather blunt tool (Riggi *et al.*, 2018). We have now come to the conclusion that different mechanisms other than, or in addition to, PIP2 changes drive TORC2 inhibition in our system. In this study, we use the 2xPH(PLC) FLARE exclusively as a generic PM marker, not as a readout of PIP2 biology. Rather, we propose that sterol redistribution and/or the biophysical impact that this has on the PM are central drivers, with TORC2 acting as a signaling node that senses and adjusts PM composition accordingly. We now clarify these arguments in the revised Discussion and have reframed our use of PalmC as a probe to explore the capacity of the PM to adapt to acute stress via dynamic lipid rearrangements.

Major issue 3:

As the authors point out, a large number of intercalated amphipaths displace sterols from their association with bilayer phospholipids. This unphysiologic mechanism can explain how PalmC causes the transient increase in the availability of plasma membrane ergosterol to the D4H probe and its subsequent removal from the plasma membrane via LAM2/4. TORC2 regulation may not be involved. In fact, the authors say that "TORC2 inhibition, and thereby Lam2/4 activation, cannot be the only trigger for PalmC induced sterol removal."

Furthermore, the subsequent recovery of plasma membrane ergosterol could simply reflect homeostatic responses independent of the components studied here.

Response:

We agree that increased free sterols in the inner leaflet likely initiate retrograde transport. Our results suggest that TORC2 inhibition facilitates this process by disinhibiting Lam2/4, allowing more efficient clearance of ergosterol from the PM (Fig. 3A, S2C). However, the process is not exclusively dependent on TORC2, and we state this explicitly. We do not observe recovery of PM ergosterol on the timescales measured, while TORC2 activity recovers, suggesting that restoration likely occurs later via biosynthetic or anterograde trafficking pathways, which are outside the scope of this study. These points are clarified in the revised Discussion.

Major issue 3a:

The data suggest that LAM2/4 mediates the return of cytoplasmic ergosterol to the plasma membrane. To my knowledge, this is a nice finding that not been reported previously and is worth confirming more directly.

Response:

We thank the reviewer for this observation but would like to clarify a misunderstanding: our data do not suggest that Lam2/4 mediates anterograde sterol transport. Our results and prior work (Gatta et al., 2015; Roelants et al., 2018) show that Lam2/4 mediate retrograde transport from the PM to the ER, and TORC2 inhibits this process. We now clarify this point in the revised manuscript, stating:

“In vivo, Lam2/4 seem to predominantly transport sterols from the PM to the ER, following the concentration gradient (Gatta et al., 2015; Jentsch et al., 2018; Tong et al., 2018).”

Major issue 4:

I agree with the authors that "It is unclear if the excess of free sterols itself is part of the inhibitory signal to TORC2..." Instead, the inhibition of TORC2 by PalmC may simply result from its artifactual aggregation of the anionic phospholipids (especially, PIP2) needed for TORC2 activity. This would not be biologically meaningful. If the authors wish to show that accessible ergosterol inhibits TORC2 activity or vice versa, they should use more direct methods. For example, neutral amphipaths that do not cause the aforementioned PalmC perturbations should still increase plasma membrane ergosterol and send it through LAM2/4 to the ER.

Response:

We now provide evidence that three orthologous treatments (hyperosmotic shock, heat shock and Arp2/3 inhibition) similarly cause sterol mobilization and, in the absence of sterol clearance from the PM, prolonged TORC2 inhibition. These results do not support the reviewer's contention that the inhibition of TORC2 by PalmC is simply resulting from its artifactual aggregation of the anionic phospholipids. Furthermore, PalmC is zwitterionic, and its interaction with anionic lipids should be somewhat limited.

In our experimental setup, neutral amphipaths did not trigger TORC2 inhibition or D4H redistribution. While this differs from prior in vitro work (Lange et al., 2009), we attribute this in part to a discrepancy to experimental setup differences, including flow chamber artifacts that we discuss in the methods section.

Importantly, only amphipaths with a charged headgroup, including zwitterionic (PalmC) and positively charged analogs, produced robust effects. A negatively charged derivative also seemed to have a minor effect on TORC2 activity and PM sterol internalization (Palmitoylglycine (Fig. 1D, Rebuttal Fig. 1). This suggests that *in vivo*, charge-based membrane perturbation is required to alter PM sterol distribution and TORC2 activity.

Major issue 5.:

The mechanistic relationship between TORC2 activity and ergosterol suggested in the title, abstract, and discussion is not secure. I agree with the concluding section of the manuscript called "Limitations of the study". It highlights the need for a better approach to the interplay between TORC2 and ergosterol.

Response:

This may have been true of the previous submission, but we now demonstrate that provoking PM stress in four orthogonal ways triggers mobilization of sterols, which left uncleared, prevents normal (re)activation of TORC2 activity. We thus conclude that free sterols, directly or more likely indirectly, inhibit TORC2. The role that TORC2 plays in sterol retrotranslocation has been demonstrated previously (Roelants et al., 2018). We believe our expanded data and clarified framework make a compelling case for a stress-adaptive role of sterol retrograde transport that is supervised and modulated—but not fully driven—by TORC2 activity.

Thus, we feel in the present version of this manuscript that the title is now justified.

Minor issue:

Based on earlier work using the reporter flipTR, the authors claim that PalmC reduces membrane tension. They should consider that this intercalated dye senses many variables including membrane tension but also lipid packing. I suspect that, by intercalating into and thereby altering the bilayer, PalmC is affecting the latter rather than the former.

Response:

We thank the reviewer for this important point regarding the multifactorial sensitivity of intercalating dyes such as Flipper-TR®, including to membrane tension and lipid packing. We respectfully note, however, that our current study does not include any new data generated using Flipper-TR®. We referred to earlier work (Riggi et al., 2018) for context, where Flipper-TR® was used as a membrane tension reporter.

We fully agree that the response of such "smart" membrane probes integrates multiple biophysical parameters—including tension, packing, and hydration—which are themselves interrelated as consequences of membrane composition (Colom et al., 2018; Ragaller et al., 2024; Torra et al., 2024). Indeed, this interconnectedness is central to our interpretation of PalmC's pleiotropic effects on the plasma membrane (PM). In our previous study, we observed that PalmC treatment not only reduced apparent PM tension (as measured by Flipper-TR®) but also increased membrane order ((Riggi et al., 2018); see laurdan GP, Fig. 6C), and here we show that it promotes the redistribution of free sterol away from the PM. Furthermore, PalmC's effect on membrane tension was supported by orthogonal in vitro data: its addition to giant unilamellar vesicles (GUVs) led to a measurable increase in membrane surface area and decreased tension, as shown by pipette aspiration ((Riggi et al., 2018), Fig. 3F). This provides complementary evidence that the membrane tension reduction is not merely an artifact of Flipper-TR® reporting.

That said, we agree with the reviewer that in the case of TORC2 inhibition or hyperactivation, the observed changes in PM tension are based solely on Flipper-TR® data, without additional orthogonal validation. To address this concern, we have revised the relevant text in the manuscript to more cautiously reflect this complexity. The revised sentence now reads:

“Consistent with this role, data generated with the lipid packing reporter dye Flipper-TR® suggest that acute chemical inhibition of TORC2 increases PM tension, while Ypk1 hyperactivation decreases it.”

This revised phrasing acknowledges both the utility and the limitations of Flipper-TR® as a probe of membrane biophysics.

Reviewer #2 Significance:

This is an interesting topic. However, use of the exogenous probe, palmitoylcarnitine, could be causing multiple changes that complicate the interpretation of the data. Reviewers #1 and #3 were much more impressed by this study than I was. I am not a yeast expert and so I may have missed or confused something. I would therefore welcome their expert feedback regarding my comments (#2). Ted Steck

Response:

Thank you for your constructive feedback.

We believe that the manuscript is now much improved, and we hope to have convinced you that the mechanisms that we've elucidated using PalmC represent a general adaptation response to physiological PM stressors.

Reviewer #3 (Evidence, reproducibility and clarity (Required)):

Reviewer comment:

The authors describe the effects of surfactant-like molecules on the plasma membrane (PM) and its associated TORC2 complex. Addition of the surfactants with a positively-charged headgroup and a hydro-carbon tail of at least 16 caused the rapid clustering of PI-4,5P2 together with PI-4P and phosphatidylserine in large membrane invaginations. The authors convincingly demonstrate that this effect of the surfactants on the PM is likely caused by a direct disturbance of the PM organization and/or lipid composition. Interestingly, upon PalmC treatment, free ergosterol of the PM was found to first concentrate in the clusters, but within <5min this ergosterol seemed to be transported into intracellular structures, causing an overall loss in free ergosterol of the PM. The authors speculate that the initial spike in free ergosterol might be the trigger for the shutdown of TORC2 signaling. The PalmC-triggered transport of free ergosterol from the PM to intracellular structures required the lipid transport proteins Lam2/4. Loss of these transporters caused a delay in TORC2 reactivation, supporting the idea that ergosterol transport out of the PM plays a role in the recovery of normal PM organization. Hyperosmotic shock mimics some of the effects observed with PalmC, but unlike PalmC treatment, TORC2 recovery after hyperosmotic shock is not dependent on Lam2/4.

The presented data are of high quality and most conclusions are well supported. However, based on the presented data the model that a PalmC-triggered increase in free ergosterol is the cause of TORC2 inactivation is not obvious to me.

The kinetics of the changes in free ergosterol levels and the changes in TORC2 activity do not match. Ergosterol is rapidly depleted after PalmC treatment (<5min) whereas TORC2 activity requires 30min to recover. Also, the hyperosmotic data on free ergosterol levels and TORC2 activity do not match. In fact, the presence of the large PM invaginations is a better predictor of TORC2 activity.

The Lam2/4 data support the idea that ergosterol transport plays a role in the TORC2 recovery, but what role this is, is not clear to me. I think the data fit better with a model in which PalmC causes low tension of the PM which in turn disrupts normal lipid organization and thus causes TORC2 to shut down, maybe not by changes in free ergosterol but by changes, for instance, in lipid raft formation (which is in part effected by ergosterol levels). The transport of ergosterol is only one mechanism that is involved in restoring PM tension and TORC2 activity. However, sensing free ergosterol alone is most likely not the mechanism explaining how TORC2 senses PM tension.

Therefore, I recommend that the model is revised (or supported by more data), reflecting the fact that free ergosterol levels do not directly correlate with the TORC2 activity, but instead might be only one of the PM parameters that regulate TORC2.

Author response:

We thank the reviewer for their thoughtful assessment and constructive suggestions. As described in more detail above, we have included in our revised version of this manuscript a variety of new data, including the sterol-internalization dependent adaptation of the PM and regulation of TORC2 during additional stresses. We think that these data vastly improve on our previous manuscript version. We have addressed each point risen by the reviewer below and revised the manuscript accordingly, including a rewritten discussion and updated model to better reflect the limitations of our current understanding of how TORC2 senses changes in the plasma membrane (PM). It is true that the appearance of PM invaginations tracks well with TORC2 inhibition, but it is not clear to us if they are upstream of this inhibition or merely another symptom of the preceding PM perturbation (PalmC-induced free sterol increase can

be observed after 10s (Fig. S2A), but PM invaginations become visible only after ~1 min – meanwhile we can observe near complete TORC2 inhibition after 30s). In this study, we are mostly interested in the role of PM sterol redistribution in stress response. Indeed we think that the role of free sterol clearance during stresses is to adapt the PM to these stresses – thus restoring PM parameters which in turn reactivates TORC2. This can be seen for hyperosmotic stress and the newly introduced PM stressors, Arp2/3 inhibition and heat shock response (Fig. 4). We have therefore softened our model and updated discussion and final figure (Fig. 5) to reflect that TORC2 likely responds to broader changes in PM organization or tension, with sterol redistribution representing one of several contributing factors rather than the sole signal.

Comment:

- If TORC2 is indeed inhibited by free ergosterol, the addition of ergosterol to the growth medium should be able to trigger similar effects as PalmC. If this detection of free ergosterol is very specific (e.g. if TORC2 has a binding pocket for ergosterol) we would expect that addition of other sterols such a cholesterol or ergosterol precursors should not inhibit TORC2.

Response:

We appreciate this suggestion and agree that testing whether exogenous ergosterol can mimic PalmC effects would help assess specificity. However, yeast do not readily take up sterols under aerobic conditions, which renders artificial sterol enrichment at the yeast PM rather difficult. We have now included additional data characterizing our Lam2/4 mutants (see below), and pharmacological sterol synthesis inhibition, showing that a depletion of free sterols from the PM correlates with lower TORC2 activity (Fig. 2D, S2C). Additionally, as suggested, we tried to probe if ergosterol directly interacts with TORC2 through a specific binding pocket, by treating a yeast strain expressing cholesterol rather than ergosterol (Souza et al., 2011) with PalmC. However, the response of TORC2 activity in these cells was very similar to that of WT cells (Rebuttal Fig. 2). In conclusion, we agree that at present we do not know mechanistically how sterols affect TORC2 activity, although it does indeed seem more likely to be through an indirect mechanism linked to changes in PM parameters. The nature of such a mechanism will be subject to further studies. We hope that the introduced changes to the manuscript adequately reflect these considerations.

Rebuttal Fig. 2: WT yeast cells which produce ergosterol as main sterol, and mutant cells which produce cholesterol instead were treated with 5 μ M PalmC, and TORC2 activity was assessed by relative phosphorylation of Ypk1 on WB. One representative experiment out of two replicates.

Comment:

- The experiment in Figure 1C is not controlled for differences in membrane intercalation of the different compounds. For instance, does C16 choline and C16 glycine accumulate at the

same rate in the PM (measure similar to experiment in Figure 1B). Maybe the positive charge at the headgroup of the surfactants increases the local concentration at the PM and therefore can explain the difference in effect on the PM.

Response:

We agree with the reviewer that the effects of the various PalmC derivatives are not directly controlled for differences in membrane intercalation. Our structure–activity screen was intended to demonstrate the general biophysical mode of action of PalmC-like compounds and to define minimal structural requirements for activity.

We now note in the manuscript that differential membrane insertion could contribute to the observed variation in efficacy, particularly in relation to tail length. While we considered this additional suggested experiment, it was ultimately judged to be outside the scope of this study due to its complexity and limited impact on the central conclusions.

A clarifying sentence has been added to the relevant results section to explicitly acknowledge this limitation:

“We did not control for differences in PM intercalation efficiency.”

We also include a discussion here to further clarify our interpretation. Prior in vitro studies have shown that while intercalation is necessary, it is not sufficient for PM perturbation. For example, palmitoyl-CoA intercalates into membranes but does not induce the same biophysical effects as PalmC (Goñi et al., 1996; Ho et al., 2002). Thus, we believe that intercalation is only part of the story, and that the intrinsic propensity of different headgroups to perturb the PM plays a key role in the disruption of PM lipid organization.

Comment:

- Are the intracellular ergosterol structures associated (or in close proximity) with lipid droplets (ergosterol being modified and delivered into a lipid droplet)?

Response:

We thank the reviewer for raising this point. We now include additional data (Fig. S2H) showing that intracellular D4H-positive structures do not reside near or colocalize with lipid droplets. The latter is not entirely unexpected as D4H does not recognize esterified sterols. However, we do observe an increase in overall LD volume following PalmC treatment, consistent with the idea that internalized PM sterols may be stored in LDs as sterol esters over time - although we did not test if this increase in LD volume is Lam2/4 dependent. This increase is mentioned in the revised results text. An increase in cellular LDs has also been recently reported during hyperosmotic shock (Phan et al., 2025).

For more attempts to identify a marker for intracellular D4H foci, see reply to reviewer 1.

Comment:

- How does the AA and DD mutations in Lam2/4 change the localization of the ergosterol sensor (before and after PalmC treatment)?

Response:

We thank the reviewer for this question, as in the course of generating these data we realized that our “inhibited” DD mutant was in fact not phosphomimetic but displayed the same D4H distribution as the “hyperactive” AA mutant, i.e. a marked inwards shift of D4H signal away from the PM to internal structures due to increased PM-ER retrograde transport

of sterols (Fig. S2C). This led us to critically re-evaluate and ultimately repeat our TORC2 activity WB experiments for PalmC treatment in *LAM2/4* mutants. In this new set of experiments, the faster TORC2 recovery after PalmC treatment in the *LAM2^{T518A} LAM4^{S401A}* mutant did unfortunately not repeat robustly. It is possible that such differences can be observed under specific conditions. Nevertheless, the improved overall quality of the Western blot data allowed us to make the observation that baseline activity was already slightly different in these strains. The Lam2/4 centered part of the results section has subsequently been updated in the manuscript:

“Using a phosphospecific antibody, we did not observe an increase in baseline TORC2 activity in *lam2Δ lam4Δ* cells, which had been previously reported by electrophoretic mobility shift (Murley *et al.*, 2017). Instead, baseline TORC2 activity was consistently slightly decreased in these cells (Fig. 2D). Ypk1, activated directly by TORC2, inhibits Lam2 and Lam4 through phosphorylation on Thr518 and Ser401, respectively (Roelants *et al.*, 2018; Topolska *et al.*, 2020). We substituted these residues with alanine, generating a strain in which Lam2/4 were no longer inhibited by phosphorylation (Roelants *et al.*, 2018). In these cells, yeGFP-D4H showed that free sterols were constitutively shifted away from the PM to intracellular structures (Fig. S2C, bottom panel). Intriguingly, in opposition to *lam2Δ lam4Δ* cells, basal TORC2 activity was increased in *LAM2^{T518A} LAM4^{S401A}* cells (Fig. 2D). This suggests that a decrease in free PM sterols stimulates TORC2 activity [...]”
“In *LAM2^{T518A} LAM4^{S401A}* cells, TORC2 activity recovers with similar kinetics as the WT (Fig. 2D, bottom blot), suggesting that Lam2/4 release from TORC2 dependent inhibition during PalmC treatment is a fast and efficient process in WT cells, not further expedited by these constitutively active Lams.”

As suggested, we also observed D4H localization in *LAM2^{T518A} LAM4^{S401A}* after PalmC treatment, and implemented these data to further demonstrate that PalmC causes an increase in the fraction of free ergosterol at the PM, which is subsequently removed:

“PalmC addition to *LAM2^{T518A} LAM4^{S401A}* cells likewise resulted first in a transient increase and then a further decrease in PM yeGFP-D4H signal (Fig. 3C, S3D).”

Comment:

- Does Lam2/4 localize to ER-PM contact sites near the large PM invaginations, which could allow for efficient transport of the free ergosterol that accumulates in these structures.

Response:

We were curious about this too, and have now added the requested data in our supplementary material and added a sentence in our results:

“Indeed, in cells expressing GFP-Lam2 we observed that PalmC induced PM invaginations often formed at sites with preexisting GFP-Lam2 foci (Fig. S2K, cyan arrow), although GFP-Lam2 foci did not always colocalize with invaginations (Fig. S2K, yellow arrow) and vice versa. “

Additionally, in the effort to characterize intracellular D4H foci during PalmC as requested by reviewer 1, we also looked at the localization of these foci relative to ER, and found that

“During early timepoints, intracellular foci are usually in close vicinity to ER (Fig. S2E)”

Reviewer #3 (Significance (Required)):

The manuscript describes the effects of small molecule surfactants on the PM organization and on TORC2 activity. This is an important set of observation that helps understanding the response of cells to environmental stressors that affect the PM. This field of study is very challenging because of the limited tools available to directly observe lipids and their movements. I consider the data and most of its interpretations of high importance, but I am not convinced of the larger model that tries to link the ergosterol data with TORC2 activity. With adjustments of the model or additional experimental support, this manuscript will be of general interest for cell biologists, especially for researchers studying membrane stress response pathways.

Response:

We thank the reviewer for highlighting the importance of studying PM stress responses and acknowledging the technical challenges involved. We hope the applied changes and additional data succeed in softening our claims about TORC2 regulation while convincing the reviewer that free sterol levels at the PM are one of several contributing factors that correlate with changes in TORC2 activity.

- Colom, A., Derivery, E., Soleimanpour, S., Tomba, C., Dal Molin, M., Sakai, N., González-Gaitán, M., Matile, S., Roux, A., 2018. A Fluorescent Membrane Tension Probe. *Nat. Chem.* 10, 1118–1125. <https://doi.org/10.1038/s41557-018-0127-3>
- Flesch, F.M., Yu, J.W., Lemmon, M.A., Burger, K.N.J., 2005. Membrane activity of the phospholipase C- δ 1 pleckstrin homology (PH) domain. *Biochem. J.* 389, 435–441. <https://doi.org/10.1042/BJ20041721>
- Gatta, A.T., Wong, L.H., Sere, Y.Y., Calderón-Noreña, D.M., Cockcroft, S., Menon, A.K., Levine, T.P., 2015. A new family of StART domain proteins at membrane contact sites has a role in ER-PM sterol transport. *eLife* 4. <https://doi.org/10.7554/eLife.07253>
- Goñi, F.M., Requero, M.A., Alonso, A., 1996. Palmitoylcarnitine, a surface-active metabolite. *FEBS Lett.* 390, 1–5. [https://doi.org/10.1016/0014-5793\(96\)00603-5](https://doi.org/10.1016/0014-5793(96)00603-5)
- Ho, J.K., Duclos, R.I., Hamilton, J.A., 2002. Interactions of acyl carnitines with model membranes: a (^{13}C) -NMR study. *J. Lipid Res.* 43, 1429–1439. <https://doi.org/10.1194/jlr.m200137-jlr200>
- Jentsch, J.-A., Kiburu, I., Pandey, K., Timme, M., Ramlall, T., Levkau, B., Wu, J., Eliezer, D., Boudker, O., Menon, A.K., 2018. Structural basis of sterol binding and transport by a yeast StArkin domain. *J. Biol. Chem.* 293, 5522–5531. <https://doi.org/10.1074/jbc.RA118.001881>
- Murley, A., Yamada, J., Niles, B.J., Toulmay, A., Prinz, W.A., Powers, T., Nunnari, J., 2017. Sterol transporters at membrane contact sites regulate TORC1 and TORC2 signaling. *J. Cell Biol.* 216, 2679–2689. <https://doi.org/10.1083/jcb.201610032>
- Otzen, D.E., Blans, K., Wang, H., Gilbert, G.E., Rasmussen, J.T., 2012. Lactadherin binds to phosphatidylserine-containing vesicles in a two-step mechanism sensitive to vesicle size and composition. *Biochim. Biophys. Acta BBA - Biomembr., Protein Folding in Membranes* 1818, 1019–1027. <https://doi.org/10.1016/j.bbamem.2011.08.032>
- Phan, J., Silva, M., Kohlmeyer, R., Ruethemann, R., Gay, L., Jorgensen, E., Babst, M., 2025. Recovery of plasma membrane tension after a hyperosmotic shock. *Mol. Biol. Cell* 36, ar45. <https://doi.org/10.1091/mbc.E24-10-0436>
- Ragaller, F., Sjule, E., Urem, Y.B., Schlegel, J., El, R., Urbancic, D., Urbancic, I., Blom, H., Sezgin, E., 2024. Quantifying Fluorescence Lifetime Responsiveness of Environment-Sensitive Probes for Membrane Fluidity Measurements. *J. Phys. Chem. B* 128, 2154–2167. <https://doi.org/10.1021/acs.jpcc.3c07006>
- Riggi, M., Niewola-Staszewska, K., Chiaruttini, N., Colom, A., Kusmider, B., Mercier, V., Soleimanpour, S., Stahl, M., Matile, S., Roux, A., Loewith, R., 2018. Decrease in plasma membrane tension triggers PtdIns(4,5)P₂ phase separation to inactivate TORC2. *Nat. Cell Biol.* 20, 1043–1051. <https://doi.org/10.1038/s41556-018-0150-z>
- Rodríguez-Escudero, I., Fernández-Acero, T., Cid, V.J., Molina, M., 2018. Heterologous mammalian Akt disrupts plasma membrane homeostasis by taking over TORC2 signaling in *Saccharomyces cerevisiae*. *Sci. Rep.* 8, 7732. <https://doi.org/10.1038/s41598-018-25717-w>
- Roelants, F.M., Chauhan, N., Muir, A., Davis, J.C., Menon, A.K., Levine, T.P., Thorner, J., 2018. TOR complex 2-regulated protein kinase Ypk1 controls sterol distribution by inhibiting StArkin domain-containing proteins located at plasma membrane-endoplasmic reticulum contact sites. *Mol. Biol. Cell* 29, 2128–2136. <https://doi.org/10.1091/mbc.E18-04-0229>
- Sakata, K.-T., Hashii, K., Yoshizawa, K., Tahara, Y.O., Yae, K., Tsuda, R., Tanaka, N., Maeda, T., Miyata, M., Tabuchi, M., 2022. Coordinated regulation of TORC2 signaling by MCC/eisosome-associated proteins, Pil1 and tetraspan membrane proteins during the stress response. *Mol. Microbiol.* 117, 1227–1244. <https://doi.org/10.1111/mmi.14903>
- Shao, C., Novakovic, V.A., Head, J.F., Seaton, B.A., Gilbert, G.E., 2008. Crystal Structure of Lactadherin C2 Domain at 1.7Å Resolution with Mutational and Computational Analyses of Its Membrane-binding Motif*. *J. Biol. Chem.* 283, 7230–7241. <https://doi.org/10.1074/jbc.M705195200>

- Shi, J., Heegaard, C.W., Rasmussen, J.T., Gilbert, G.E., 2004. Lactadherin binds selectively to membranes containing phosphatidyl-L-serine and increased curvature. *Biochim. Biophys. Acta* 1667, 82–90. <https://doi.org/10.1016/j.bbamem.2004.09.006>
- Souza, C.M., Schwabe, T.M.E., Pichler, H., Ploier, B., Leitner, E., Guan, X.L., Wenk, M.R., Riezman, I., Riezman, H., 2011. A stable yeast strain efficiently producing cholesterol instead of ergosterol is functional for tryptophan uptake, but not weak organic acid resistance. *Metab. Eng.* 13, 555–569. <https://doi.org/10.1016/j.ymben.2011.06.006>
- Stefan, C.J., Audhya, A., Emr, S.D., 2002. The yeast synaptojanin-like proteins control the cellular distribution of phosphatidylinositol (4,5)-bisphosphate. *Mol. Biol. Cell* 13, 542–557. <https://doi.org/10.1091/mbc.01-10-0476>
- Tong, J., Manik, M.K., Im, Y.J., 2018. Structural basis of sterol recognition and nonvesicular transport by lipid transfer proteins anchored at membrane contact sites. *Proc. Natl. Acad. Sci.* 115, E856–E865. <https://doi.org/10.1073/pnas.1719709115>
- Topolska, M., Roelants, F.M., Si, E.P., Thorner, J., 2020. TORC2-Dependent Ypk1-Mediated Phosphorylation of Lam2/Ltc4 Disrupts Its Association with the β -Propeller Protein Laf1 at Endoplasmic Reticulum-Plasma Membrane Contact Sites in the Yeast *Saccharomyces cerevisiae*. *Biomolecules* 10, 1598. <https://doi.org/10.3390/biom10121598>
- Torra, J., Campelo, F., Garcia-Parajo, M.F., 2024. Tensing Flipper: Photosensitized Manipulation of Membrane Tension, Lipid Phase Separation, and Raft Protein Sorting in Biological Membranes. *J. Am. Chem. Soc.* 146, 24114–24124. <https://doi.org/10.1021/jacs.4c08580>
- Uekama, N., Aoki, T., Maruoka, T., Kurisu, S., Hatakeyama, A., Yamaguchi, S., Okada, M., Yagisawa, H., Nishimura, K., Tuzi, S., 2009. Influence of membrane curvature on the structure of the membrane-associated pleckstrin homology domain of phospholipase C- δ 1. *Biochim. Biophys. Acta BBA - Biomembr.* 1788, 2575–2583. <https://doi.org/10.1016/j.bbamem.2009.10.009>

Dear Aurélien,

We have now received re-review reports from the three referees that originally appraised your manuscript for Review Commons, which I have included below. As you will see, you have addressed their concerns satisfactorily; however, I would like you to consider addressing the remaining relevant points of Referee #2 in the discussion section. Before I can finally accept the manuscript, there are some remaining editorial points which need to be addressed. In this regard would you please:

- remove the figures from the main manuscript file,
- acknowledge funding from the Swiss National Science Foundation the Canton of Geneva with grant number: project 310030_207754 in the manuscript,
- limit the number of keywords to five, listing them below the abstract,
- in the reference section, limit longer author lists to the first 10 authors + et al.,
- change the title of the conflict of interests statement to "Disclosure and competing interests statement",
- remove the AC/CrediT section from the text,
- correct callouts for Figures S1-S5 to Appendix Figure S1-S5,
- complete the appropriate checklists,
- upload figures as as individual, high-resolution files with the legends placed in the main manuscript below the references,
- convert the appendix file to PDF format; the title page should contain "Appendix for A dynamic retrograde sterol transport - TORC2 feedback loop adapts the plasma membrane during stress" and a table of contents with the page numbers for the listed items; nomenclature should be Appendix Figure Sx and Appendix Table Sx throughout ms and Appendix PDF,
- include a Reagents and Tools table,
- contact contact@embojournal.org for instructions about deposition of source data, and if necessary include a data availability statement in the manuscript,
- refer to figure S2N in the manuscript,
- define box plots in terms of minima, maxima, percentile in the legend of figure S2B,
- define the nature of n in the legends of figures 1B, D, E; 2A, D, E; 4B, D, F; S1 A-E; S2 J, N; S4 A,
- define error bars in the legends of figures S2 K, N,
- rename movies as Movie EV1-EV13 with the appropriate callouts and uploaded individually and the corresponding legends zipped with each movie file, and
- correct the section order as follows: Title page - Abstract - Keywords - Introduction - Results - Discussion - Methods - Data Availability - Acknowledgements - Disclosure and Competing Interests Statement - References - Figure Legends - Table(s) - Expanded View Figure Legends.

We include a synopsis of the paper (see <http://emboj.embojournal.org/>). Please provide me with a general summary image, a two sentence statement and 3-5 bullet points that capture the key findings of the paper.

I am looking forward to receiving your revised manuscript.

EMBO Press is an editorially independent publishing platform for the development of EMBO scientific publications.

Best wishes,

William

William Teale, PhD
Editor
The EMBO Journal
w.teale@embojournal.org

See also figure legend guidelines: <https://www.embojournal.org/page/journal/14602075/authorguide#figureformat>

- a point-by-point response to the referees' comments, with a detailed description of the changes made (as a word file).
 - a word file of the manuscript text.
 - individual production quality figure files (one file per figure)
 - a complete author checklist, which you can download from our author guidelines (<https://www.embopress.org/page/journal/14602075/authorguide>).
 - Expanded View files (replacing Supplementary Information)
- Please see out instructions to authors
<https://www.embopress.org/page/journal/14602075/authorguide#expandedview>
- a Reagents and Tools Table as part of the Methods section, which can be downloaded from our author guidelines (<https://www.embopress.org/page/journal/14602075/authorguide#structuredmethods>)

We realize that it is difficult to revise to a specific deadline. In the interest of protecting the conceptual advance provided by the work, we recommend a revision within 3 months (9th Nov 2025). Please discuss the revision progress ahead of this time with the editor if you require more time to complete the revisions. Use the link below to submit your revision:

Referee #1:

All my concerns were adequately addressed.

Referee #2:

071625 Re-review of Tettamanti et al. for Embo J.

From Reviewer #2:

The authors have responded constructively and expertly with an improved manuscript. While I still have some reservations, the data speak for themselves and the authors are entitled to their interpretations. Thus, this manuscript is suitable for publication by the Embo J. Some last thoughts:

1. I am embarrassed to have carelessly taken PalmC to be a cationic rather than a zwitterionic amphiphile in my initial review. Apologies.
2. You reference literature concluding that alkylphospholipids can mimic the PalmC effect. In other systems, alkylphospholipids sequester sterols and do not "mobilize" them. See Rios-Marco, Carrasco, et al.
3. The following distinction might be useful: The accessible sterol in the plasma membrane is a small fraction of the total. In contrast, bilayer physical properties such as membrane tension - or whatever the probe is reporting - presumably reflect the major, complexed portion of the sterol. The marginal accessible sterol has apparently not been quantitated in yeast, but it seems to amount to ~2% in animal cells. Thus, the accessible fraction does not reflect or influence bulk "membrane biophysics" except as a homeostatic signal, as you say.
4. For what it's worth, I fear that the Doktorova bilayer "model" is not secure. Caution advised.
5. A fine point in this regard: At different points, you invoke two different mechanisms for the primary action of PalmC: Lange (2009) versus Doktorova (2025). The former predicts that the sterol in the outer and inner leaflets will rise and fall synchronously due to its displacement from complexes and rapid transit bilayer equilibration. The latter predicts that the outer leaflet sterol will fall as the inner leaflet rises due to "area displacement" by PalmC. Have you tracked the accessible outer plasma membrane leaflet sterol with a soluble sterol probe like PFO or AIdD4? (But according to Doktorova, there should be a large amount of probe-accessible sterol in the unperturbed outer leaflet; but they did not check this. Not finding this large outer leaflet sterol pool would undermine that mechanism.)
6. Does PalmC have any other effect than activating PM ergosterol? As you know well, many other intercalated amphiphiles displace and activate complexed plasma membrane cholesterol (Lange, 2009). Have you tried long chain fatty acids? They should activate PM ergosterol (at least transiently) and help you to distinguish that function from possible other effects of PalmC intercalation.
8. You are also well aware that the level of accessible sterol varies at a steep threshold in plasma membrane sterol concentration and can change acutely with physiologic and experimental conditions. This is tricky and could underlie some of the irregularity in experiments.

Referee #3:

The authors addressed all my major concerns and therefore I support the publication of this manuscript.

Referee #1:

All my concerns were adequately addressed.

Referee #2:

071625 Re-review of Tettamanti et al. for Embo J.

From Reviewer #2:

The authors have responded constructively and expertly with an improved manuscript. While I still have some reservations, the data speak for themselves and the authors are entitled to their interpretations. Thus, this manuscript is suitable for publication by the Embo J. Some last thoughts:

1. I am embarrassed to have carelessly taken PalmC to be a cationic rather than a zwitterionic amphiphile in my initial review. Apologies.
2. You reference literature concluding that alkylphospholipids can mimic the PalmC effect. In other systems, alkylphospholipids sequester sterols and do not "mobilize" them. See Rios-Marco, Carrasco, et al.

2. Response:

We thank the reviewer for pointing out this discrepancy. The relevance of alkylphospholipids in tumour therapy arises from the observation that their cytotoxic effects appear to selectively target tumour cells, such as the HepG2 tumour cells studied by Carrasco and colleagues (Jiménez-López *et al*, 2006; Ríos-Marco *et al*, 2017). This selectivity is likely due to differences in plasma membrane composition, organisation, and lipid metabolism in these cells, potentially causing differential effects (Fei *et al*, 2023). We agree with the reviewer that researchers working with these membrane-perturbing amphiphiles should be aware that their effects on membranes are pleiotropic and may vary across systems and between different analogues. However, we believe that a detailed discussion of this is beyond the scope of our study, particularly since we do not provide direct lipid observations that would clarify the precise nature of amphiphile-induced lipid perturbations. To better represent our citation of the yeast alkylphospholipid literature, we now explicitly state in the text that sterol relocalization with an alkylphospholipid was observed in budding yeast:

"For instance, alkylphospholipid analogs (Fei et al, 2023) [...] relocalize sterols in budding yeast (Zaremborg et al, 2005)."

3. The following distinction might be useful: The accessible sterol in the plasma membrane is a small fraction of the total. In contrast, bilayer physical properties such as membrane tension - or whatever the probe is reporting - presumably reflect the major, complexed portion of the sterol. The marginal accessible sterol has apparently not been quantitated in yeast, but it seems to amount to ~2% in animal cells. Thus, the accessible fraction does not reflect or influence bulk "membrane biophysics" except as a homeostatic signal, as you say.
4. For what it's worth, I fear that the Doktorova bilayer "model" is not secure. Caution advised.
5. A fine point in this regard: At different points, you invoke two different mechanisms for the

primary action of PalmC: Lange (2009) versus Doktorova (2025). The former predicts that the sterol in the outer and inner leaflets will rise and fall synchronously due to its displacement from complexes and rapid transit bilayer equilibration. The latter predicts that the outer leaflet sterol will fall as the inner leaflet rises due to "area displacement" by PalmC. Have you tracked the accessible outer plasma membrane leaflet sterol with a soluble sterol probe like PFO or AloD4? (But according to Doktorova, there should be a large amount of probe-accessible sterol in the unperturbed outer leaflet; but they did not check this. Not finding this large outer leaflet sterol pool would undermine that mechanism.)

6. Does PalmC have any other effect than activating PM ergosterol? As you know well, many other intercalated amphiphiles displace and activate complexed plasma membrane cholesterol (Lange, 2009). Have you tried long chain fatty acids? They should activate PM ergosterol (at least transiently) and help you to distinguish that function from possible other effects of PalmC intercalation.

3.-6. Response:

Following all our observations with PalmC, we believe that its effects on the PM are pleiotropic. We did indeed try one long chain fatty acid in the SAR screen (C16/Palmitate), however it did not have any effect in our system, as was the case for amphipaths with uncharged headgroups in our experiments. This suggests that the charged headgroup of PalmC plays a critical role in plasma membrane perturbation, extending beyond the effects reported for small amphipaths in Lange 2009. We propose that the induced perturbation arises from a combination of sterol displacement (Lange, 2009), charge dependent perturbation (Requero *et al*, 1995) and "area displacement" by PalmC-like molecules, followed by equilibration involving sterols, as discussed in Doktorova 2025. We specifically did not address sterol leaflet asymmetry in our manuscript, since (1) this is a big and very controversial topic (Steck & Lange, 2018) and we have no relevant data, and (2) sterol distribution might differ across organisms or even cell types. Indeed, a recent study using DHE quenching in budding yeast reported an inverse transbilayer distribution of sterol compared to Doktorova 2025, with more ergosterol in the inner leaflet.

Although we did not use D4H or other protein-based probes to directly assess sterols in the outer leaflet, we performed control experiments with Filipin staining. Filipin, a fluorescent antifungal that binds unesterified sterols, can stain the outer plasma membrane leaflet. Consistent with Zarembeg *et al.* (2005), we observed that PalmC treatment caused a strong increase in Filipin staining at the plasma membrane, which we interpreted as a change in membrane organisation leading to a massive increase of free sterol. After 60 minutes—when TORC2 activity had recovered in our western blot experiments— PM Filipin staining diminished to below baseline levels – consistent with internalization (Rebuttal 2 Figure 1). Zarembeg *et al.* (2005) further showed that in fixed permeabilized cells, Filipin accumulated in intracellular foci upon treatment with edelfosine. The discrepancy in timescales between Filipin and D4H likely reflects differences in the membrane leaflet examined and the staining mechanisms involved. While these Filipin experiments provided a useful proof of concept, we chose not to include them in the manuscript due to technical limitations: Filipin's rapid toxicity to yeast, its requirement for UV imaging, and the low resolution and overall quality of the data generated with the only compatible system available.

In summary, we believe that our data support the view that PalmC perturbs the plasma membrane through multiple, overlapping mechanisms. It seems possible that a significant

fraction of sequestered sterol gets dislodged and can now be internalized to adapt the PM to a functional state.

We agree that one sentence in the last subchapter of our discussion was worded in a way that might imply that sterol mobilization is a result of bilayer imbalance rather than a potentially simultaneous effect of PalmC. We have slightly tweaked this sentence to reflect this explanation:

“Thus, we propose that PalmC accumulates in the outer leaflet, where it frees a fraction of sequestered sterols due to its membrane perturbing properties (Lange et al, 2009; Requero et al, 1995), and disrupts the lipid balance with the inner leaflet, which is, similarly to the mammalian cell model (Doktorova et al, 2025), rectified by sterol mobilization, flipping and internalization (Fig. 5B).”

Rebuttal 2 Figure 1: Living budding yeast cells stained with Filipin complex for membrane sterol. Yeast cells were grown to OD600nm 0.6-0.8, and 5 μ M PalmC was added to the liquid culture. 1 ml culture aliquots were washed once with 50 mM potassium phosphate buffer pH 5.5 (1.5 min 5000 xg), resuspended in 1 ml of the same buffer containing 5 μ g/ml filipin complex (Sigma-Aldrich, readymade stock: 5 mg/ml in DMSO). Cells were incubated for 4-5 min in the dark at RT, washed again with the same buffer, and mounted onto high precision glass slides, before imaging with a Leica DMI8 widefield microscope, equipped with a 100x1.4 oil objective, a Leica A4 UV filter cube (ex. 425-475 nm, beamsplitter 400 nm, em. 435-485nm) and an Evolve 512 EMCCD camera with a pixel size of 16 μ m, using LAS X (3.3.3.16958) software.

8. You are also well aware that the level of accessible sterol varies at a steep threshold in plasma membrane sterol concentration and can change acutely with physiologic and experimental conditions. This is tricky and could underlie some of the irregularity in experiments.

This is very true – and the experimental variance is reflected in the slight variance of some of the baseline D4H distributions in untreated cells between repeat experiments (see e.g. Fig. 2B and source data). We took particular care in keeping experimental conditions constant between different experiments, and treating cells gently when preparing slides so as to not

cause mechanical stress.

Referee #3:

The authors addressed all my major concerns and therefore I support the publication of this manuscript.

Dear Aurélien,

I am pleased to inform you that your manuscript has been accepted for publication in the EMBO Journal.

Congratulations to you and your team!

Best wishes,

William

William Teale, PhD
Editor
The EMBO Journal
w.teale@embojournal.org
